# ADAPTIVE GAUSSIAN EXPANSION FOR ON-THE-FLY CATEGORY DISCOVERY

**Chunming Li**[1], **Shidong Wang**[2], **Haofeng Zhang**[1]*

[1]Nanjing University of Science and Technology, Nanjing, China
[2]Newcastle University, Newcastle upon Tyne, United Kingdom
[1]{chunmingli,zhanghf}@njust.edu.cn, [2]shidong.wang@newcastle.ac.uk

## ABSTRACT

On-the-Fly Category Discovery (OCD) aims to address the limitations of transductive learning and closed-set prediction in category discovery tasks by enabling real-time classification of potential future categories using prior knowledge. Existing OCD approaches typically rely on hash-based encodings that map features into low-dimensional hash spaces and directly classify test samples using these encodings. Despite efforts to mitigate the sensitivity of hash functions during testing, these methods still suffer from severe overestimation of the number of categories. In this work, we thoroughly analyze the practical limitations of current OCD methods and formally identify a performance lower bound for the task. Based on this insight, we reformulate OCD into two sub-tasks: Open-Set Recognition and an *Fully Novel OCD* setting. For all samples, we employ a soft class thresholding strategy to directly detect known classes, which significantly enhances the deployment feasibility of OCD to downstream tasks. For outlier samples, we propose Adaptive Gaussian Expansion (AGE), a dynamic category discovery method that models the Probability Density Functions (PDF) of different classes to uncover potential novel categories in real time. Extensive experiments across multiple datasets demonstrate that our method achieves state-of-the-art performance. Code is available at: https://github.com/Ashengl/AGE

## 1 INTRODUCTION

As models continue to scale, many now exhibit recognition capabilities that surpass human performance (He et al., 2016; 2022; Dosovitskiy et al., 2021). While existing models benefit from large-scale datasets to achieve accurate classification of known categories, they are typically specialized for closed-set prediction. Consequently, when faced with new datasets requiring annotation, these models struggle to leverage prior knowledge of known categories to effectively model novel ones. To address this limitation, researchers have introduced a new task setting known as Category Discovery. Category Discovery is commonly divided into

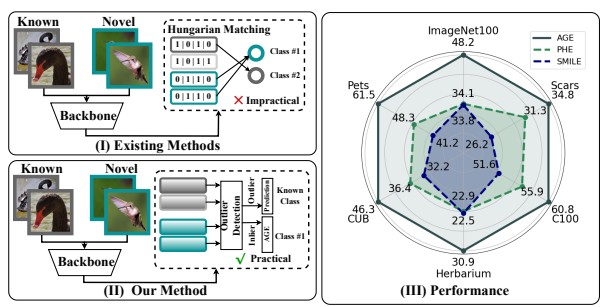

Figure 1: Overview of our method versus prior approaches. (I) Existing methods predict all samples before category matching, which is impractical. (II) Our method directly classifies known samples and infers novel ones via the AGE framework. (III) Experiments show consistent improvements across datasets.

Novel Category Discovery (NCD) (Rizve et al., 2022) and Generalized Category Discovery (GCD) (Vaze et al., 2022a; Wen et al., 2023), distinguished by whether the provided new unlabeled dataset includes samples from known categories.

---

*Corresponding author.

Although the concept of Category Discovery effectively alleviates the constraints of closed-set prediction, its reliance on transductive learning paradigms or assumptions of label overlap between training and test sets still distances it from the ideal open-world discovery scenario. Consequently, a more challenging task, On-the-fly Category Discovery (OCD) (Du et al., 2023), has emerged. Unlike GCD and NCD, OCD relaxes the requirements on the training set, demanding models be trained only on a partially labeled dataset. Without assuming predefined categories, the model must leverage information acquired during training to infer the categories of test samples in real-time, even when test samples include novel categories.

A common strategy involves decomposing sample feature embeddings into sign and magnitude components, using the sign as a proxy for category identity. However, this approach suffers from a critical limitation in its high sensitivity, as the way the model defines a category can differ significantly from human semantic understanding, potentially relying on irrelevant features to identify new categories. Moreover, hash-based encoding in OCD poses a significant challenge for downstream tasks. Even for well-established categories with ample training data, the model continues to depend on hash-based detection to identify the most likely cluster corresponding to an old class, rather than directly and accurately recognizing the known category.

According to the OCD evaluation metric based on Hungarian matching, which explores all possible assignments to find the maximum performance across all classes, we argue that model performance has a lower bound. Given a trained closed-set classifier, assuming no novel categories exist, the model can already achieve respectable performance under the OCD metric (see Table 1 Close-set I). If we assume that novel categories do exist and performance is maximized through the matching process, an unintuitive result emerges: relying solely on the closed-set classifier can yield performance comparable to state-of-the-art methods (see Table 1 Close-set II). We therefore define this performance as a lower bound for OCD models.

Table 1: Performance with close-set classifier.

| Method | ImageNet-100 | | | CUB-200 | | | Herbarium19 | | |
|---|---|---|---|---|---|---|---|---|---|
| | All | Old | New | All | Old | New | All | Old | New |
| SLC | 32.9 | 86.6 | 5.2 | 28.6 | 44.0 | 20.9 | 14.9 | 27.4 | 8.1 |
| MLDG | 30.6 | 72.3 | 9.7 | 29.5 | 48.4 | 20.1 | 20.8 | 36.7 | 12.3 |
| RankStat | 31.1 | 73.3 | 9.8 | 21.2 | 26.9 | 18.4 | 13.8 | 20.6 | 10.2 |
| WTA | 30.8 | 72.9 | 9.7 | 21.9 | 26.9 | 19.4 | 14.6 | 21.2 | 11.1 |
| SMILE | 33.8 | 74.2 | 13.5 | 32.2 | 50.9 | 22.9 | 22.9 | 39.3 | 14.1 |
| PHE | 34.1 | 80.6 | 10.8 | 36.4 | 55.8 | 27.0 | 22.5 | 38.5 | 14.0 |
| Close-set I | 32.0 | 96.1 | 0 | 28.5 | 85.6 | 0 | 22.2 | 66.5 | 0 |
| Close-set II | 38.7 | 85.3 | 15.3 | 35.5 | 52.2 | 27.1 | 26.1 | 52.7 | 11.2 |

Given this identified lower bound, we argue that existing methods do not fully or effectively exploit the information in the labeled data, highlighting the need for more robust approaches to novel category identification. Under the established evaluation criteria, hash encoding proves ineffective for accurately classifying seen categories and is therefore not a robust method for real-time novel category detection. In this paper, we propose to reframe the OCD task as a combination of Open-Set Recognition and real-time novel category identification. Given that different labeled datasets may impose different criteria for defining novelty, we further decompose the real-time identification task according to the granularity of the available labeled data.

Inspired by the Chinese Restaurant Process (CRP) in Dirichlet Process Gaussian Mixture Model (DPGMM) (Blei & Jordan, 2006), we propose an online incremental clustering and classification method called Adaptive Gaussian Expansion (AGE), which simultaneously addresses known category recognition and novel category discovery. Specifically, all sample features are first projected into a lower-dimensional space to eliminate redundancy, reduce collinearity, and improve computational efficiency. Then, a stable covariance matrix is estimated for each known class subset using the Ledoit–Wolf shrinkage estimator (Ledoit & Wolf, 2004), and a global covariance matrix is constructed as a prior. During inference, since test samples are entirely unknown, we adopt a greedy strategy that makes locally optimal decisions based on the available information. Each new sample is matched against existing clusters using the Mahalanobis distance (De Maesschalck et al., 2000) and Gaussian Probability Density Function (PDF). If its maximum posterior probability falls within an existing cluster, the mean and covariance of that cluster are incrementally updated according to the Dirichlet process concentration parameter; otherwise, the sample is treated as the seed of a new cluster, and its parameters are initialized accordingly. This greedy, posterior-based cluster assignment rule adaptively forms new clusters during streaming inference without requiring a predefined number of categories. Experimental results demonstrate that our method achieves state-of-the-art performance across multiple datasets, with an average accuracy improvement of 10% over all categories.

Our contributions can be summarized as follows:

- We establish a theoretical lower bound for OCD performance, revealing inherent limitations of prior approaches and motivating a reformulation of the task.
- We reformulate the OCD task as a dual subproblem, enabling explicit disentanglement of known and novel class inference via a statistically grounded, class-specific thresholding mechanism.
- We propose AGE for OCD, achieving state-of-the-art results across benchmarks.

## 2 RELATED WORKS

### 2.1 OPEN SET RECOGNITION

Open Set Recognition (OSR) aims to address the challenge of encountering unknown classes during testing that were not present in the training phase. Unlike traditional closed-set classification, which assumes that all test samples belong to known categories, OSR explicitly models the uncertainty and incompleteness of the training data. OSR methods are generally divided into two main approaches: generative models (Ge et al., 2017; Jo et al., 2018; Neal et al., 2018) and discriminative models (Bendale & Boult, 2016; Chen et al., 2020; Guo et al., 2021). While generative models aim to model the underlying data distribution, discriminative models focus on learning decision boundaries between known classes. Recent work (Kasarla et al., 2022) highlights that enforcing a prior that encourages well-separated embeddings remains essential for achieving strong performance in open-set recognition. Open Set Recognition (OSR) focuses on identifying unknown classes during the testing phase, while OCD goes a step further by attempting to subdivide these unknown classes into distinct and meaningful categories.

### 2.2 CATEGORY DISCOVERY

Novel Category Discovery (NCD) aims to identify novel classes from unlabeled data with limited labeled examples, unlike semi-supervised learning which assumes shared class space. This task was first formally introduced by Deep Transfer Clustering (DTC) (Han et al., 2019), which combines unsupervised clustering (DEC) with prototype-based classification. Autonovel (Han et al., 2020) adopts self-supervised pretraining along with a ranking strategy to generate pseudo-labels, enabling joint training with labeled samples. NCL (Zhong et al., 2021) leverages contrastive learning to enhance feature discriminability, while UNO (Fini et al., 2021) presents a unified architecture that simultaneously learns from both labeled and novel categories. For example, DCCL (Pu et al., 2023) employs an alternating optimization strategy to extract visual concepts and learn their representations; PromptCAL (Zhang et al., 2023) introduces prompt-based learning; SimGCD (Wen et al., 2023) integrates contrastive objectives with parametric classification; and SPTNet (Wang et al., 2024) proposes a two-stage iterative framework that incorporates spatial prompt tuning. More recently, to address the constraint in NCD that all class data must be seen during training, OCD (Du et al., 2023) has been introduced. OCD removes the dependency on training-set access, enabling real-time, streaming inference on entirely unseen, unlabeled data. Recently PHE (Zheng et al., 2024) distinguishes known and unknown classes via prototype hypersphere estimation, while Diff-GRE (Liu et al., 2025) leverages multimodal and generative models for unknown class discovery.

## 3 METHODOLOGY

### 3.1 SETTING

In OCD tasks, the data typically consist of two sets: a labeled support set $D_S = \{(\boldsymbol{x}_i, \boldsymbol{y}_i)\}_{i=1}^N \subseteq \mathcal{X} \times \mathcal{Y}_S$, containing a subset of labeled samples, and an unlabeled query set $D_Q = \{(\boldsymbol{x}_i, \boldsymbol{y}_i)\}_{i=1}^M \subseteq \mathcal{X} \times \mathcal{Y}_Q$ encountered during inference, where the label space satisfies $\mathcal{Y}_S \subseteq \mathcal{Y}_Q$. We define the set of novel classes as $\mathcal{Y}_N = \mathcal{Y}_Q \setminus \mathcal{Y}_S$. In traditional supervised learning, $\mathcal{Y}_N = \emptyset$, whereas in OCD tasks, $\mathcal{Y}_N \neq \emptyset$ and typically contains multiple unseen classes. During the training phase, only the support set $D_S$ is available. Unlike most existing approaches, we further split $D_S$ into a validation set $D_V$.

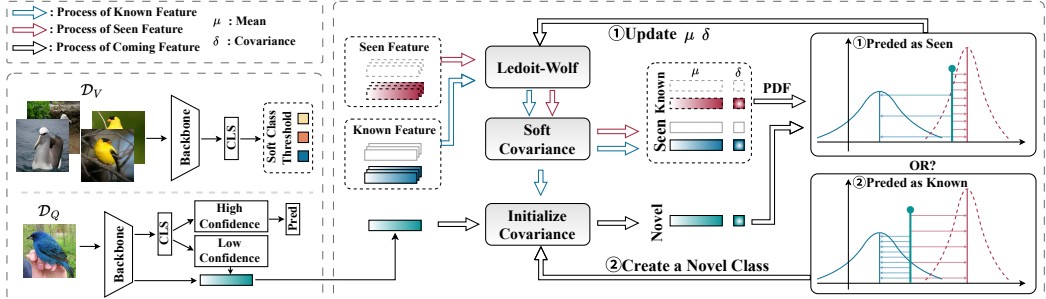

Figure 2: Overview of the AGE framework. Incoming features are first classified by a closed-set module; low-confidence samples enter the AGE module, which uses adaptive covariance estimates to update existing class distributions or create new components for novel category discovery.

During the testing phase, the model processes incoming samples in a streaming fashion and receives ground-truth labels in real-time.

## 3.2 METHOD OVERVIEW

We decompose our model into an encoder $\mathcal{E}$, an MLP projection head $\mathcal{H}$, and a parameterized classifier $\mathcal{C} = [\{\phi_i\}_{i=1}^{|\mathcal{Y}_S|}]$, where each classifier weight vector $\phi_i$ is $\ell_2$-normalized and and does not include a bias term. As outlined in the introduction, the core challenge of the OCD problem lies in effectively distinguishing between known-class samples and anomalous instances within $D_Q$. Within the proposed framework, the model is initially trained on the support set. During inference, class-specific thresholds are estimated using a held-out validation set. For each incoming query sample, the model first conducts closed-set classification via a normalized classifier and then determines whether the sample is anomalous based on the corresponding threshold. Samples identified as anomalies are subsequently handled by our proposed AGE, which enables flexible and incremental discovery of new classes.

## 3.3 TRAINING

Unlike existing methods such as SMILE and PHE, which typically decouple features into sign and magnitude components and assign the sign as a unique hash code, our approach performs discriminative analysis directly in the feature space. Consequently, we do not employ a feature-decoupled projection head. Instead, we train the model on the support set $D_S$ using a combination of cross-entropy loss and supervised contrastive learning.

Formally, given a mini-batch of samples $B_S = \{(x_i, y_i)\}$, the supervised contrastive loss is computed as:

$$\mathcal{L}_{\text{sup}} = \frac{1}{|B_S|} \sum_{i \in B_S} \frac{1}{|\mathcal{P}(i)|} \sum_{p \in \mathcal{P}(i)} -\log \frac{\exp(\boldsymbol{z}_i \cdot \boldsymbol{z}_p / \tau)}{\sum_{i \neq a} \exp(\boldsymbol{z}_i \cdot \boldsymbol{z}_a / \tau)}, \tag{1}$$

where $\boldsymbol{z}_i = \mathcal{H}(\mathcal{E}(\boldsymbol{x}_i)) \in \mathbb{R}^d$ is the $\ell_2$-normalized projection of sample $x_i$ and $\mathcal{P}_i$ indexes images that hold the same label as $\boldsymbol{x}_i$ in $B_S$.

The standard supervised cross-entropy loss is defined as:

$$\mathcal{L}_{\text{cls}} = -\frac{1}{|B_S|} \sum_{i \in B_S} \boldsymbol{y}_i \log \boldsymbol{p}_i, \tag{2}$$

where $\boldsymbol{p}_i = \mathcal{C}(\boldsymbol{z}_i)$. The total training objective is the weighted sum of the two losses:

$$\mathcal{L} = \mathcal{L}_{\text{cls}} + \mathcal{L}_{\text{sup}}. \tag{3}$$

## 3.4 INFERENCE

As previously mentioned, in our constructed dataset, we split a portion of the support set to form a validation set, which is then used to estimate the threshold for detecting anomalous samples during the testing phase. Unlike global thresholding strategies that compute confidence thresholds based

on the overall distribution of all samples, we propose a soft thresholding method. For any sample $\boldsymbol{x}_i \in D_V$, we define its confidence as $\gamma_i = \max(\text{Softmax}(\boldsymbol{p}_i))$. For all samples of class $k$, we define the mean and standard deviation of their confidence scores as follows:

$$\mu_k = \frac{1}{N_k} \sum_{i=1}^{N_k} \gamma_i^{(k)}, \tag{4}$$

$$\sigma_k = 0.8 \sqrt{\frac{1}{N_k - 1} \sum_{i=1}^{N_k} \left(\gamma_i^{(k)} - \mu_k\right)^2}. \tag{5}$$

The final threshold is obtained by averaging these class-wise thresholds:

$$\tau_c = \beta * (\mu_c - \sigma_c) + (1 - \beta) * \frac{1}{|\mathcal{Y}_S|} \sum_{k \in \mathcal{Y}_S} \mu_k - \sigma_k. \tag{6}$$

Here, $\beta$ represents the fusion ratio between class-specific and global thresholds. While a global threshold may overlook certain low-confidence classes, class-specific thresholds can become excessively noisy when the validation set is small.

The proposed method effectively mitigates distortion caused by class imbalance and inter-class covariance by modeling intra-class statistical properties. It enhances the discriminative performance on known classes while maintaining strong anomaly detection capabilities. Most existing OOD detection methods rely on metrics such as OSCR (Dhamija et al., 2018) and H-score (Fu et al., 2020), requiring manual threshold adjustment based on specific scenarios or validation sets. As a concrete embodiment of anomaly detection in real-world scenarios, the AGE-based OCD task holds significant value in advancing the deployment and practical application of anomaly detection methods.

We apply the predefined threshold to all samples in the query set $D_Q$. If a sample $\boldsymbol{x}_i$ has a confidence score above the threshold, it is classified as belonging to a known class. Otherwise, its $\ell_2$-normalized feature representation $\boldsymbol{f} = \mathcal{E}(\boldsymbol{x}_i)$ is extracted and fed into the proposed real-time category discovery framework AGE. Ideally, the model should accurately distinguish all known-class samples from anomalous ones. This assumption serves as a crucial foundation for the validity of the subsequent category discovery process.

**Ledoit-Wolf** The Ledoit-Wolf shrinkage estimator improves covariance estimation in high-dimensional settings with limited samples by shrinking the empirical covariance matrix toward a more stable target. Given the small validation set and the incremental accumulation of samples from newly discovered classes during inference, we apply this method to enhance the robustness of covariance estimation. For any set of features $\{\boldsymbol{f}_i\}_{i=1}^{N_k}$ belonging to the same class $k$, we first compute the sample mean and the unbiased estimate of the covariance matrix:

$$\bar{\boldsymbol{f}} = \frac{1}{N_k} \sum_{i=1}^{n} \boldsymbol{f}_i \in \mathbb{R}^d, \qquad \boldsymbol{S} = \frac{1}{N_k - 1} \sum_{i=1}^{N_k} (\boldsymbol{f}_i - \bar{\boldsymbol{f}})(\boldsymbol{f}_i - \bar{\boldsymbol{f}})^\top \in \mathbb{R}^{d \times d}. \tag{7}$$

The Ledoit-Wolf covariance estimator is defined as

$$\hat{\Sigma}_{\text{LW}}(\boldsymbol{S}) = (1 - \lambda^*)\boldsymbol{S} + \lambda^* \boldsymbol{F}, \tag{8}$$

where $\boldsymbol{F} = \frac{1}{d}\text{tr}(\boldsymbol{S})\boldsymbol{I}$ is the shrinkage target, with $\boldsymbol{I}$ denoting the identity matrix. The coefficient $\lambda^*$ is the optimal shrinkage parameter that balances the contribution between the empirical covariance matrix $\boldsymbol{S}$ and the target matrix $\boldsymbol{F}$. It is computed as:

$$\lambda^* = \frac{\mathbb{E}\left[\|\boldsymbol{S} - \Sigma\|_F^2\right]}{\mathbb{E}\left[\|\boldsymbol{F} - \Sigma\|_F^2\right] + \mathbb{E}\left[\|\boldsymbol{S} - \Sigma\|_F^2\right]}. \tag{9}$$

However, since the true covariance matrix $\Sigma$ is not accessible, Ledoit and Wolf proposed an unbiased estimator. In practice, the optimal shrinkage coefficient is computed as:

$$\lambda^* = \frac{\frac{1}{n} \sum_{i=1}^{n} \left\|(\boldsymbol{x}_i - \bar{\boldsymbol{x}})(\boldsymbol{x}_i - \bar{\boldsymbol{x}})^\top - \boldsymbol{S}\right\|_F^2}{\|\boldsymbol{S} - \boldsymbol{F}\|_F^2}. \tag{10}$$

For simplicity, during real-time inference on $D_Q$, we denote the Ledoit-Wolf covariance estimate for class $k$ as $\Sigma_{\text{LW}}^k$. In the remainder of the paper, we refer to $\Sigma_{\text{LW}}^k$ simply as "variance".

**Adaptive Gaussian Expansion**  Assuming we have extracted features $\boldsymbol{F}_S = \bigcup_{c \in \mathcal{Y}_S} \{[\boldsymbol{f}_j^{(c)}]\}_{j=1}^{N_c}$ from all samples in the validation set, we first compute the mean feature vector and covariance matrix for all features, denoted as $\{\boldsymbol{\mu}_1, \boldsymbol{\mu}_2, \cdots, \boldsymbol{\mu}_{|\mathcal{Y}_S|}\}$ and $\boldsymbol{\Sigma}_{\text{all}}$. To mitigate the imbalance caused by long-tailed distributions in covariance estimation, we adopt a smoothed covariance strategy to estimate the covariance matrices for different classes:

$$\boldsymbol{\Sigma}_k = (1 - \alpha_k)\boldsymbol{\Sigma}_{\text{LW}}^k + \alpha_k \boldsymbol{\Sigma}_{\text{all}} + \varepsilon \boldsymbol{I}, \tag{11}$$

where $\alpha_k = \frac{s}{N_k + s}$ denotes soft prior weight, and is introduced to prevent the covariance estimation from being excessively biased when the sample size is small, which could lead to a significant performance drop. The term $\varepsilon \boldsymbol{I}$ serves as a numerical stability factor. As $\varepsilon$ increases, the covariance approaches the identity matrix, causing the model to degenerate into Euclidean distance-based model. In practice, we introduce a fixed-size sliding window to prevent memory consumption from growing linearly with the number of samples. The window operates in a first-in, first-out manner, such that once the number of stored samples exceeds a predefined limit, the earliest samples are discarded.

During the inference phase, the model continuously discovers new categories. At any given time, suppose we have a set $\boldsymbol{F}_P = \bigcup_{c \in \mathcal{Y}_P} \{[\boldsymbol{f}_j^{(c)}]\}_{j=1}^{N_P}$ of features from different discovered novel categories, along with their corresponding means and covariances, which are computed according to Eq. 11. Here, $N_P$ denotes the number of novel categories discovered so far. For any feature $\boldsymbol{f}_{od}$ identified as an outlier, we initialize covariance as $\boldsymbol{\Sigma}_{\text{od}} = \boldsymbol{\Sigma}_{\text{all}}$.

Most conventional methods rely on classifiers to obtain the logit of a sample, which only reflects its relative proximity to known classes and fails to capture the intra-class distribution characteristics. In contrast, classification using a multivariate Gaussian PDF explicitly estimates the mean vector and covariance matrix for each class, enabling direct computation of the generative likelihood of the sample under that class. This allows for a more precise characterization of the known class distributions, thereby significantly enhancing the generalization ability and the quality of uncertainty estimation.

Under ideal conditions, for a given feature $\boldsymbol{f}_{od}$ that does not belong to any known category, the task model can be simplified to distinguishing whether it belongs to a seen or an unseen class. We argue that the feature distribution of known classes can serve as a reasonable background prior for modeling unseen classes. To theoretically support this intuition, we present the following results.

Lemma 1 shows that, after whitening, features from semantically related unseen classes $C$ are geometrically closer to the training classes $A$ than to unrelated seen-but-target classes $B$.

**Lemma 1.** *Assume that the feature vectors of all classes follow a multivariate Gaussian distribution with a shared covariance matrix $\boldsymbol{\Sigma}$. After applying a whitening transformation such that $\boldsymbol{\Sigma} = \boldsymbol{I}$, the Mahalanobis distance reduces to the Euclidean distance. Let $\boldsymbol{\mu}_A, \boldsymbol{\mu}_B, \boldsymbol{\mu}_C$ denote the class-wise mean features of groups A, B, and C, respectively. Define the squared distances:*

$$\boldsymbol{d}_A = \|\boldsymbol{\mu}_C - \boldsymbol{\mu}_A\|^2, \quad \boldsymbol{d}_B = \|\boldsymbol{\mu}_C - \boldsymbol{\mu}_B\|^2.$$

*Then, under the following assumptions: 1). The encoder is trained solely on group A and learns to capture its semantic structure. 2). The features of group C align more strongly with the semantic subspace spanned by group A than by group B, i.e., $\boldsymbol{\mu}_C^\top \mathbb{E}[\boldsymbol{\mu}_A] > \boldsymbol{\mu}_C^\top \mathbb{E}[\boldsymbol{\mu}_B]$. 3). The distribution of means in group A is at least as dispersed as that in group B: $\mathbb{E}\|\boldsymbol{\mu}_A\|^2 \geq \mathbb{E}\|\boldsymbol{\mu}_B\|^2$.*

*It follows that the expected distance satisfies $\mathbb{E}[\boldsymbol{d}_A] < \mathbb{E}[\boldsymbol{d}_B]$, i.e., the expected distance between group C and the known training group A is smaller than that between C and the unrelated group B.*

Building on this, Proposition 1 further demonstrates that when the Gaussian distribution of training classes $A$ is adopted as a background model for "non-$B$" samples, and if the distribution of unseen class $C$ is closer to $A$, the model can more accurately classify $C$ samples as "non-$B$" in a binary decision setting (i.e., $B$ vs. non-$B$), thereby providing theoretical support for new class discovery.

**Proposition 1.** *Assume that samples from classes A, B, and C in the feature space follow multivariate Gaussian distributions with class means $\boldsymbol{\mu}_A$, $\boldsymbol{\mu}_B$, and $\boldsymbol{\mu}_C$, and a shared covariance matrix $\boldsymbol{\Sigma}$. The label spaces of A, B, and C are disjoint. Suppose that class A is used as a negative (non-B)*

*background model for open-set recognition, and samples are classified as A (i.e., not B) if*

$$(\boldsymbol{x} - \boldsymbol{\mu}_A)^\top \boldsymbol{\Sigma}^{-1}(\boldsymbol{x} - \boldsymbol{\mu}_A) < (\boldsymbol{x} - \boldsymbol{\mu}_B)^\top \Sigma^{-1}(\boldsymbol{x} - \boldsymbol{\mu}_B).$$

*Then, if the Mahalanobis distance between $\mu_C$ and $\mu_A$ is smaller than that between $\mu_C$ and $\mu_B$:*

$$(\boldsymbol{\mu}_C - \boldsymbol{\mu}_A)^\top \boldsymbol{\Sigma}^{-1}(\boldsymbol{\mu}_C - \boldsymbol{\mu}_A) < (\boldsymbol{\mu}_C - \boldsymbol{\mu}_B)^\top \boldsymbol{\Sigma}^{-1}(\boldsymbol{\mu}_C - \boldsymbol{\mu}_B),$$

*a sample drawn from class $C$ is more likely to be classified as $A$ (non-$B$) than as $B$ under this rule.*

Under the whitening assumption of the covariance matrix, the Mahalanobis distance reduces to the Euclidean distance. Proposition provides an intuitive rationale for the OCD task, suggesting that the distribution of known classes can serve as a reasonable prior for modeling unknown classes. In practice, AGE initializes new categories using the global prior $\Sigma_{\text{all}}$ and applies smoothing, which in turn approximately preserves the shared covariance assumption.

For an outlier sample $\boldsymbol{f}_{od}$, its probability density is computed as:

$$\boldsymbol{p}_k = \frac{1}{(2\pi)^{d/2}|\boldsymbol{\Sigma}|^{1/2}} \exp\left(-\frac{1}{2}(\boldsymbol{f}_{od} - \boldsymbol{\mu}_k)^\top \boldsymbol{\Sigma}_k^{-1}(\boldsymbol{f}_{od} - \boldsymbol{\mu}_k)\right). \tag{12}$$

In high-dimensional settings, raw PDF values can be extremely small, causing underflow errors. Using the logarithm transforms these tiny values into sums, preventing precision issues. Thus, we convert the PDF to its log form:

$$\log \boldsymbol{p}_k = -\tfrac{1}{2}\left(d\log(2\pi) + \log|\boldsymbol{\Sigma}_k| + (\boldsymbol{f}_{od} - \boldsymbol{\mu_k})^\top \boldsymbol{\Sigma}_k^{-1}(\boldsymbol{f}_{od} - \boldsymbol{\mu_k})\right). \tag{13}$$

For each sample detected as an outlier, AGE first evaluates its likelihood under the multivariate Gaussian distributions of all known classes. If the sample is unlikely to belong to any known class, it is treated as a potential new category. Its feature is then incorporated into the set of discovered classes, with the initial covariance set based on the global background covariance and the class mean initialized accordingly.

To avoid numerical instability during PDF computation, we apply Principal Component Analysis (PCA) dimensionality reduction to the features during inference.

## 4 EXPERIMENTS

The proposed AGE was evaluated on several benchmark datasets for image recognition tasks. These include the CIFAR-100 (Krizhevsky et al., 2009) dataset, the ImageNet-100 dataset (Krizhevsky et al., 2012)), and the recently introduced Semantic Shift Benchmark datasets (Vaze et al., 2022b) which consist of the CUB-200 (Reed et al., 2016), and Stanford-Cars (Krause et al., 2013). Additionally, the more challenging Herbarium 2019 fine-grained classification dataset (Tan et al., 2019) was also used for evaluation. To enable a more comprehensive comparison, similar to PHE, we also evaluated on Pets (Parkhi et al., 2012) and iNaturalist (Van Horn et al., 2018), which includes the categories Fungi, Arachnida, Animalia, and Mollusca.

### 4.1 IMPLEMENTATION DETAILS

The backbone used in our experiments was ViT-B/16 (Dosovitskiy et al., 2021), pre-trained on the ImageNet dataset using the DINO self-supervised learning framework (Caron et al., 2021). We used consistent training and testing settings across all datasets, with a batch size of 128, a learning rate of 0.005, and a weight decay of 5e-5. The model was trained for 100 epochs. In the covariance smoothing prior, the soft parameter $s$ was set to 2, represented the relative importance of the prior covariance in the subsequent estimation of class-specific covariances. While in threshold smoothing, $\beta$ was set to 0.5. The feature dimensionality is reduced to 42 using PCA. For validation, 20% of the training data was reserved. The sliding-window size used for covariance estimation is set to 20. We discuss the effects of these hyperparameters on model performance in the ablation study and Appendix.

We compared our approach with state-of-the-art methods such as DiffGRE (Liu et al., 2025), PHE (Zheng et al., 2024), SMILE (Du et al., 2023), as well as adaptive approaches including SLC (Hartigan, 1975), MLDG (Li et al., 2018), RankStat (Han et al., 2020), and WTA (Jia et al., 2021).

Table 2: Comparison with the State of the Art methods. The best results are marked in **bold**.

| Method | CIFAR-100 | | | ImageNet-100 | | | CUB-200 | | | Scars | | | Herbarium19 | | |
|---|---|---|---|---|---|---|---|---|---|---|---|---|---|---|---|
| | All | Old | New | All | Old | New | All | Old | New | All | Old | New | All | Old | New |
| SLC | 44.4 | 59.0 | 15.1 | 32.9 | 86.6 | 5.2 | 28.6 | 44.0 | 20.9 | 14.0 | 23.0 | 9.7 | 14.9 | 27.4 | 8.1 |
| MLDG | 50.6 | 61.0 | 29.8 | 30.6 | 72.3 | 9.7 | 29.5 | 48.4 | 20.1 | 24.0 | 41.6 | 15.4 | 20.8 | 36.7 | 12.3 |
| RankStat | 35.0 | 44.0 | 17.0 | 31.1 | 73.3 | 9.8 | 21.2 | 26.9 | 18.4 | 14.8 | 19.9 | 12.3 | 13.8 | 20.6 | 10.2 |
| WTA | 40.8 | 52.9 | 16.7 | 30.8 | 72.9 | 9.7 | 21.9 | 26.9 | 19.4 | 17.1 | 24.4 | 13.6 | 14.6 | 21.2 | 11.1 |
| SMILE | 51.6 | 61.6 | 31.7 | 33.8 | 74.2 | 13.5 | 32.2 | 50.9 | 22.9 | 26.2 | 46.7 | 16.3 | 22.9 | 39.3 | 14.1 |
| PHE | 55.9 | 70.0 | 27.7 | 34.1 | 80.6 | 10.8 | 36.4 | 55.8 | 27.0 | 31.3 | 61.9 | 16.8 | 22.5 | 38.5 | 14.0 |
| DiffGRE | - | - | - | - | - | - | 42.5 | 54.4 | 36.5 | 27.7 | 48.1 | 17.8 | - | - | - |
| AGE (Ours) | **60.8** | **75.8** | **30.7** | **48.2** | **87.1** | **28.6** | **46.3** | **59.8** | **39.4** | **34.8** | **62.7** | **21.3** | **30.9** | **54.7** | **18.1** |

Table 3: Comparison with the State of the Art methods. The best results are marked in **bold**.

| Method | Fungi | | | Arachnida | | | Animalia | | | Mollusca | | | Pets | | | Food101 | | |
|---|---|---|---|---|---|---|---|---|---|---|---|---|---|---|---|---|---|---|
| | All | Old | New | All | Old | New | All | Old | New | All | Old | New | All | Old | New | All | Old | New |
| SLC | 27.7 | 60.0 | 13.4 | 25.4 | 44.6 | 11.4 | 32.4 | 61.9 | 19.3 | 31.1 | 59.8 | 15.0 | 35.5 | 41.3 | 33.1 | 20.9 | 48.6 | 6.8 |
| RankStat | 23.8 | 50.5 | 12.0 | 26.6 | 51.0 | 10.0 | 31.4 | 54.9 | 21.6 | 29.3 | 55.2 | 15.5 | 33.2 | 42.3 | 28.4 | 22.3 | 50.7 | 7.8 |
| WTA | 27.5 | 65.6 | 12.0 | 28.1 | 55.5 | 10.9 | 33.4 | 59.8 | 22.4 | 30.3 | 55.4 | 17.0 | 35.2 | 46.3 | 29.3 | 18.2 | 40.5 | 6.1 |
| SMILE | 29.3 | 64.6 | 13.6 | 29.9 | 57.9 | 12.2 | 35.9 | 49.4 | 30.3 | 33.3 | 44.5 | 27.2 | 41.2 | 42.1 | 40.7 | 24.0 | 54.6 | 8.4 |
| PHE | 31.4 | **67.9** | 15.2 | 37.0 | **75.7** | 12.6 | 40.3 | 55.7 | 31.8 | 39.9 | 65.0 | 26.5 | 48.3 | 53.8 | 45.4 | 29.1 | 64.7 | **11.1** |
| DiffGRE | - | - | - | **47.7** | 76.6 | **29.4** | 43.5 | 63.2 | **35.3** | 42.6 | 62.0 | **32.3** | 49.6 | 50.1 | 49.3 | - | - | - |
| AGE (Ours) | **40.0** | 66.1 | **28.5** | 42.7 | 73.7 | 23.3 | **44.1** | **68.3** | 34.2 | **43.0** | **70.0** | 28.7 | **61.5** | **66.5** | **58.8** | **30.5** | **70.0** | 10.4 |

Specifically, SLC is an incremental clustering algorithm that sequentially processes data, assigning samples to existing clusters or creating new ones based on a distance threshold. RankStat determines class by ranking features and using the top-3 ranked features to define class descriptors. WTA divides the feature embeddings into three groups and generates predictions based on the index of the maximum activation within each group. All experiments were conducted using an NVIDIA GeForce RTX 4090 GPU.

## 4.2 EVALUATION PROTOCOLS

Following prior work such as PHE and OCD, we adopted only the Strict-Hungarian protocol as the evaluation metric (Vaze et al., 2022a), which can be formulated as:

$$ACC = \max_{p \in \mathcal{P}(\hat{\mathcal{Y}}_Q)} \left( \frac{1}{|D_Q|} \sum_{i=1}^{|D_Q|} \mathbb{I}(y_i = p(\hat{y}_i)) \right), \tag{14}$$

where $\mathcal{P}(\hat{\mathcal{Y}}_Q)$ defines the optimal matching between predicted and ground-truth labels for the test samples. This matching is obtained using the Hungarian algorithm, which finds the permutation that minimizes the total mismatch between predictions and true labels.

## 4.3 RESULTS

Here we present a comprehensive comparison of various methods across different datasets, highlighting the performance of our proposed approach. As shown in Table 2, on two representative coarse-grained datasets, CIFAR-100 and ImageNet-100, our method achieves an average improvement of 9.5% in overall accuracy compared to PHE (Zheng et al., 2024), demonstrating its strong advantage in general category discrimination. For fine-grained datasets commonly used in GCD tasks, including CUB-200, Scars, Herbarium19, and Pets, our method achieves an average improvement of 8.8% in overall accuracy compared with PHE, further validating its generalization ability in fine-grained scenarios. Notably, compared to DiffGRE (Liu et al., 2025), which leverages multimodal and generative approaches, AGE still achieves performance that is nearly comparable.

Importantly, these performance gains are primarily attributed to the superior ability of our method to recognize novel categories. Across the six datasets mentioned above, our method improves the accuracy on novel categories by an average of 16.7%. Result highlights the effectiveness of open-set detection mechanism in identifying emerging classes and demonstrates the ability of the proposed AGE module to leverage information from known classes to accurately discover novel ones.

Table 4: Effects of each component.

| Component | | | | CUB-200 | | | ImageNet-100 | | |
|---|---|---|---|---|---|---|---|---|---|
| OSR | PDF | LW | SC | All | Old | New | All | Old | New |
| ✓ | ✓ | ✗ | ✗ | 26.2 | 57.2 | 10.7 | 35.2 | 75.5 | 30.1 |
| ✓ | ✓ | ✓ | ✗ | 27.0 | 58.1 | 11.2 | 35.8 | 76.8 | 30.6 |
| ✓ | ✓ | ✗ | ✓ | 45.0 | 57.8 | 38.6 | 47.8 | 87.1 | 28.1 |
| ✗ | ✓ | ✓ | ✓ | 38.8 | 28.6 | 43.7 | 20.9 | 0.1 | 31.1 |
| ✓ | ✓ | ✓ | ✓ | 46.3 | 59.8 | 39.4 | 48.2 | 87.1 | 28.6 |

Table 5: Comparison between different dimensionality reduction methods.

| Methods | CUB-200 | | | ImageNet-100 | | |
|---|---|---|---|---|---|---|
| | All | Old | New | All | Old | New |
| PCA | 46.3 | 59.8 | 39.4 | 48.2 | 87.1 | 28.6 |
| LDA | 46.1 | 61.6 | 38.3 | 49.1 | 78.1 | 27.2 |
| ICA | 46.4 | 63.0 | 38.1 | 44.1 | 85.3 | 23.4 |
| RP | 40.1 | 61.9 | 29.2 | 52.2 | 87.1 | 34.7 |
| PLS | 45.4 | 67.9 | 34.2 | 45.0 | 87.1 | 23.9 |
| NCA | 45.2 | 62.8 | 36.4 | 48.7 | 89.0 | 28.4 |

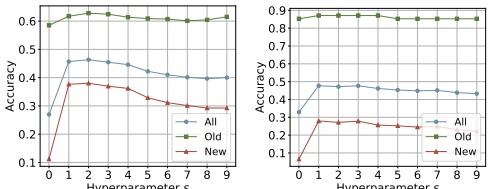

Figure 3: Performance with different $s$.

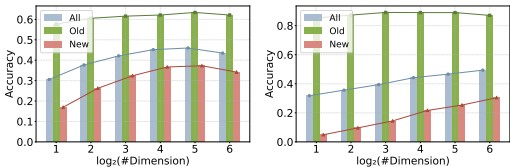

Figure 4: Performance with different dimensions.

To further verify the robustness of our framework in complex real-world settings, we conducted experiments on the more challenging iNaturalist dataset, with results presented in Table 3. Compared with PHE, our method achieves an average improvement of 5.3% in overall accuracy, with particularly strong performance on the Fungi dataset where the gain reaches 8.6%. These results further confirm the generalization capability and stability of our approach across diverse and large-scale category spaces.

## 5 ABLATION STUDY

**Effects of Components** To investigate the impact of each component in the proposed AGE framework, we conducted a series of ablation studies to systematically evaluate their contributions to the novel category discovery task. As shown in Table 4, the Soft Covariance plays a critical role. In the early stages of real-time inference, the number of samples per class is insufficient, resulting in highly unstable covariance estimation. This instability adversely affects the computation of PDF, potentially causing model instability or numerical explosion. Fig. 3 presents our investigation and analysis of the impact of soft covariance on model performance, showing that the model achieves optimal performance when $s = 2$. The incorporation of Ledoit-Wolf variance estimation offers a more robust covariance estimate, enhancing the capability to discover novel categories. When the OSR component is removed, the performance degrades significantly, as the known classes are treated as background, preventing the accurate identification of previously seen classes. This indirectly validates the effectiveness of leveraging known categories as background information to facilitate the discovery of novel classes.

**Effects of Dimensionality Reduction** Since high-dimensional density estimation is unstable due to degraded covariance estimation, we apply PCA to reduce feature dimensionality. Table 5 compares the performance of several dimensionality reduction methods, including Linear Discriminant Analysis (LDA), Independent Component Analysis (ICA), Random Projection (RP), Partial Least Squares (PLS), and Neighborhood Components Analysis (NCA). On ImageNet, RP better preserves global structure and inter-class distances due to coarse category granularity. On the fine-grained CUB-200 dataset, however, RP may distort subtle local features, leading to poorer performance.

In Fig. 4, we investigate the impact of feature dimensionality. Since numerical instability occurs in PDF computation when the dimensionality exceeds 128, we set the horizontal axis maximum to 64. As the dimensionality increases, the accuracy on ImageNet-100 steadily improves, whereas CUB-200 shows a rise-then-fall trend. This discrepancy mainly arises from the granularity differences between the two datasets: ImageNet-100 contains coarser categories, where higher dimensions help preserve more informative features; in contrast, CUB-200, being a fine-grained dataset, may suffer from increased noise or overfitting in high dimensions, leading to degraded performance.

**Effects of Numerical Stabilization Term** $\varepsilon$   To evaluate the role of the numerical stabilization term $\varepsilon \boldsymbol{I}$ in covariance estimation, we conducted a series of ablation studies by varying the value of $\varepsilon$ and analyzing its impact on final performance (as shown in Table 6). Without this stabilization term, the computation of the PDF often suffers from numerical instability due to the singularity of the covariance matrix. A properly chosen stabilization term helps suppress singularity and enhances numerical stability, particularly when the sample size is small. However, an excessively large $\varepsilon$ causes the covariance matrix

Table 6: Performance with different numerical stabilization term $\varepsilon$.

|  | CUB-200 | | | ImageNet-100 | | |
|---|---|---|---|---|---|---|
| $\varepsilon$ | All | Old | New | All | Old | New |
| 1e-7 | 45.9 | 57.2 | 40.3 | 47.6 | 87.1 | 27.8 |
| 1e-5 | 46.3 | 59.8 | 39.4 | 48.2 | 87.1 | 28.6 |
| 1e-4 | 45.8 | 62.6 | 37.4 | 47.1 | 87.1 | 27.0 |
| 1e-3 | 43.4 | 61.7 | 34.2 | 45.3 | 87.1 | 24.2 |
| 1e-2 | 30.9 | 58.8 | 17.0 | 32.7 | 85.3 | 6.2 |

to become overly diagonal-dominant, resulting in oversmoothing of the class-specific covariance structures and reducing inter-class discriminability. Consequently, the model fails to effectively capture correlations among features. As $\varepsilon$ increases, AGE gradually degenerates into SLC, which relies solely on feature distances for classification. When $\varepsilon = 1\text{e-}2$, the performance closely approaches that of SLC.

**Effects of Different OSR Methods**   We adopt MSP combined with the proposed soft-thresholding strategy as our OSR approach. In Table 7, we report the Open Set Classification Accuracy (OSAcc) and Out-of-Distribution Accuracy (OODAcc). We further investigate additional OSR methods, including Energy Score, MSP, and MSP+ for confidence estimation, and compare their AUROC and FPR@TPR95. The results indicate that using

Table 7: Performance with different OSR methods.

|  | CUB-200 | | Pets | |
|---|---|---|---|---|
|  | FPR@TPR95 | AUROC | FPR@TPR95 | AUROC |
| MSP | 76.9 | 78.0 | 63.9 | 82.3 |
| Energy Score | 77.7 | 77.9 | 61.5 | 84.3 |
| MSP+ | 76.1 | 78.4 | 73.3 | 80.0 |
|  | OSAcc | OODAcc | OSAcc | OODAcc |
| MSP+Soft | 75.4 | 74.8 | 70.1 | 60.9 |

FPR95 induce a strong bias toward known classes, particularly on fine-grained datasets. MSP+ under sharpened logits becomes even more challenging for distinguishing OOD samples. It is worth noting that conventional OSR evaluation metrics are typically based on an optimal threshold, which ultimately requires selecting a reasonable threshold. The soft-thresholding strategy proposed in this work represents an effective attempt to apply various OSR methods in practice.

**Effects of Noises**   To evaluate the potential impact of old-class noise on Gaussian estimation and the subsequent new-class discovery process, we include an additional ablation study focusing on noise injection. Specifically, we compare new-class prediction performance under three settings: (1) completely masking all old-class samples, (2) keeping the original dataset distribution unchanged, and (3) injecting simulated old-class noise into the estimation procedure. The noise samples are synthesized using the old-class mean and an amplified version of the global variance to intentionally increase their pertur-

Table 8: Performance with different Noises. New: Accuracy on New Classes, Est.: Estimated Number of Classes

|  | CUB-200 | | Pets | | Scars | |
|---|---|---|---|---|---|---|
|  | New | Est. | New | Est. | New | Est. |
| (1) | 43.5 | 329 | 64.6 | 93 | 25.2 | 429 |
| (2) | 43.9 | 401 | 64.2 | 147 | 25.1 | 634 |
| (3) | 43.2 | 452 | 78.0 | 169 | 73.3 | 697 |

bation strength. As shown in Table 8, the accuracy differences across the three settings are minimal, indicating that AGE is highly robust to old-class noise and does not suffer noticeable degradation even when a small amount of misrouted samples is introduced.

# 6   CONCLUSION

In this paper, we propose a novel framework called Adaptive Gaussian Expansion to address key limitations of existing On-the-fly Category Discovery methods. By decomposing the OCD task into a combination of sub-tasks and incorporating a dynamic probabilistic model, AGE enables robust and real-time adaptation to emerging categories. Unlike previous approaches, the proposed framework allows the model to directly predict old classes as their corresponding ground-truth labels, offering a more practical solution. Extensive experiments on diverse datasets demonstrate that AGE consistently outperforms prior methods, particularly in identifying novel categories.

ACKNOWLEDGMENTS

This work was supported in part by the National Natural Science Foundation of China under the Grants No. 62371235 and No. U25A20444, in part by the Key Research and Development Plan of Jiangsu Province under Grant No. BE2023008-2.

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

# A APPENDIX

## A.1 DETAILS OF DATASETS

For the experimental evaluation, the datasets were partitioned into labeled and unlabeled subsets. Specifically, for the CIFAR-100 dataset, 80% of the data was allocated to the labeled set $D_S$, while for the remaining datasets, this ratio was set to 50%. The labeled data was sampled from the labeled classes, and the unlabeled data $D_Q$ was formed by merging the remaining instances. As mentioned earlier, we further split 20% of the training samples as a validation set to assist with anomaly detection. Details of the datasets are provided in Table 9 and Table 10.

Table 9: Statistical comparison of data partitions.

|  |  | CIFAR-100 | ImageNet-100 | CUB-200 | Stanford-Cars | Herbarium19 |
|---|---|---|---|---|---|---|
| Labeled | $|\mathcal{Y}_S|$ | 80 | 50 | 100 | 98 | 341 |
|  | $|D_S|$ | 16k | 25.5k | 1.2k | 1.6k | 1.6k |
|  | $|\mathcal{Y}_V|$ | 80 | 50 | 100 | 98 | 341 |
|  | $|D_V|$ | 4k | 6.4k | 0.3k | 0.4k | 7.2k |
| Unlabeled | $|\mathcal{Y}_Q|$ | 100 | 100 | 200 | 196 | 683 |
|  | $|D_Q|$ | 30k | 95.3k | 4.5k | 6.1k | 25.4k |

Table 10: Statistical comparison of data partitions.

|  |  | Fungi | Arachnida | Animalia | Mollusca | Pets | Food101 |
|---|---|---|---|---|---|---|---|
| Labeled | $|\mathcal{Y}_S|$ | 61 | 28 | 39 | 47 | 341 | 51 |
|  | $|D_S|$ | 1.5k | 1.3k | 1.2k | 1.6k | 1.9k | 15.3k |
|  | $|\mathcal{Y}_V|$ | 61 | 61 | 39 | 47 | 341 | 51 |
|  | $|D_V|$ | 0.3k | 0.3k | 0.3k | 0.4k | 0.5k | 3.8k |
| Unlabeled | $|\mathcal{Y}_Q|$ | 121 | 56 | 77 | 93 | 683 | 101 |
|  | $|D_Q|$ | 5.8k | 4.3k | 5.1k | 6.1k | 7.0k | 56.6k |

## A.2 EFFECTS OF VALIDATION SET PROPORTION

Regarding the proportion of validation samples, a larger validation set helps estimate a more accurate covariance matrix, while a larger training set is beneficial for learning better features. In Table 11, we compare the impact of different validation set proportions on performance. For the fine-grained CUB-200 dataset, insufficient training samples lead to a drop in accuracy due to its limited sample size. In contrast, since the Pets dataset has fewer categories, the validation split has a smaller effect on performance.

Table 11: Performance with different validation set proportion.

|  | CUB-200 | | | Pets | | |
|---|---|---|---|---|---|---|
|  | All | Old | New | All | Old | New |
| 20% | 46.3 | 59.8 | 39.4 | 61.5 | 66.5 | 58.8 |
| 30% | 45.6 | 57.6 | 39.5 | 59.0 | 66.8 | 54.8 |
| 40% | 43.9 | 56.6 | 37.5 | 61.2 | 67.6 | 57.8 |
| 50% | 42.6 | 55.9 | 35.9 | 62.3 | 67.6 | 59.6 |
| 60% | 42.2 | 54.1 | 36.2 | 61.1 | 72.2 | 55.3 |

### A.3 PROOF OF LEMMA AND PROPOSITION

*Proof of Lemma 1.* Since the feature space is whitened, we have $\boldsymbol{\Sigma} = \boldsymbol{I}$ and thus the Mahalanobis distance becomes the squared Euclidean distance. Expanding the squared distance between the means in the whitened space yields:

$$\|\boldsymbol{\mu}_C - \boldsymbol{\mu}_A\|^2 = \|\boldsymbol{\mu}_C\|^2 + \|\boldsymbol{\mu}_A\|^2 - 2\,\boldsymbol{\mu}_C^\top\boldsymbol{\mu}_A,$$
$$\|\boldsymbol{\mu}_C - \boldsymbol{\mu}_B\|^2 = \|\boldsymbol{\mu}_C\|^2 + \|\boldsymbol{\mu}_B\|^2 - 2\,\boldsymbol{\mu}_C^\top\boldsymbol{\mu}_B.$$

Taking expectations over class means in groups $A$ and $B$ respectively:

$$\mathbb{E}[\boldsymbol{d}_A] = \|\boldsymbol{\mu}_C\|^2 + \mathbb{E}\|\boldsymbol{\mu}_A\|^2 - 2\,\boldsymbol{\mu}_C^\top\mathbb{E}[\boldsymbol{\mu}_A],$$
$$\mathbb{E}[\boldsymbol{d}_B] = \|\boldsymbol{\mu}_C\|^2 + \mathbb{E}\|\boldsymbol{\mu}_B\|^2 - 2\,\boldsymbol{\mu}_C^\top\mathbb{E}[\boldsymbol{\mu}_B].$$

Subtracting the two gives:

$$\mathbb{E}[\boldsymbol{d}_B] - \mathbb{E}[\boldsymbol{d}_A] = \left(\mathbb{E}\|\boldsymbol{\mu}_B\|^2 - \mathbb{E}\|\boldsymbol{\mu}_A\|^2\right) - 2\,\boldsymbol{\mu}_C^\top\left(\mathbb{E}[\boldsymbol{\mu}_B] - \mathbb{E}[\boldsymbol{\mu}_A]\right).$$

Under the stated assumptions:

$$\boldsymbol{\mu}_C^\top\left(\mathbb{E}[\boldsymbol{\mu}_A] - \mathbb{E}[\boldsymbol{\mu}_B]\right) > 0, \quad \mathbb{E}\|\boldsymbol{\mu}_A\|^2 - \mathbb{E}\|\boldsymbol{\mu}_B\|^2 \geq 0,$$

which implies that

$$\mathbb{E}[\boldsymbol{d}_B] - \mathbb{E}[\boldsymbol{d}_A] > 0,$$

and thus

$$\mathbb{E}[\boldsymbol{d}_A] < \mathbb{E}[\boldsymbol{d}_B].$$

$\square$

*Proof of Proposition 1.* Let $\boldsymbol{X} \sim \mathcal{N}(\boldsymbol{\mu}_C, \boldsymbol{\Sigma})$ be a sample from class $C$. The log-likelihoods under Gaussian distributions with means $\boldsymbol{\mu}_A$ and $\boldsymbol{\mu}_B$ (and shared covariance $\boldsymbol{\Sigma}$) are:

$$\log p_A(\boldsymbol{X}) = -\frac{1}{2}(\boldsymbol{X} - \boldsymbol{\mu}_A)^\top\boldsymbol{\Sigma}^{-1}(\boldsymbol{X} - \boldsymbol{\mu}_A) + \text{const},$$
$$\log p_B(\boldsymbol{X}) = -\frac{1}{2}(\boldsymbol{X} - \boldsymbol{\mu}_B)^\top\boldsymbol{\Sigma}^{-1}(\boldsymbol{X} - \boldsymbol{\mu}_B) + \text{const}.$$

Hence, the classification rule reduces to comparing Mahalanobis distances:

$$\boldsymbol{d}_A = (\boldsymbol{X} - \boldsymbol{\mu}_A)^\top\boldsymbol{\Sigma}^{-1}(\boldsymbol{X} - \boldsymbol{\mu}_A),$$
$$\boldsymbol{d}_B = (\boldsymbol{X} - \boldsymbol{\mu}_B)^\top\boldsymbol{\Sigma}^{-1}(\boldsymbol{X} - \boldsymbol{\mu}_B).$$

Let $\boldsymbol{Y} = \boldsymbol{\Sigma}^{-1/2}(\boldsymbol{X} - \boldsymbol{\mu}_C)$, so that $\boldsymbol{Y} \sim \mathcal{N}(0, I)$. Define:

$$a = \boldsymbol{\Sigma}^{-1/2}(\boldsymbol{\mu}_C - \boldsymbol{\mu}_A), \quad b = \boldsymbol{\Sigma}^{-1/2}(\boldsymbol{\mu}_C - \boldsymbol{\mu}_B).$$

Then we can rewrite:

$$\boldsymbol{d}_A = \|\boldsymbol{Y} + a\|^2, \quad \boldsymbol{d}_B = \|\boldsymbol{Y} + b\|^2.$$

Thus, the condition $\boldsymbol{d}_A < \boldsymbol{d}_B$ becomes:

$$\|\boldsymbol{Y} + a\|^2 < \|\boldsymbol{Y} + b\|^2 \iff 2(b - a)^\top\boldsymbol{Y} > \|a\|^2 - \|b\|^2.$$

Since $\boldsymbol{Y} \sim \mathcal{N}(0, I)$, the projection $(b - a)^\top\boldsymbol{Y}$ is a scalar Gaussian random variable with zero mean and symmetric distribution. If $\|a\|^2 < \|b\|^2$, then the threshold on the right-hand side is negative, and we have:

$$P(\boldsymbol{d}_A < \boldsymbol{d}_B) = P\left((b - a)^\top\boldsymbol{Y} > \frac{1}{2}(\|a\|^2 - \|b\|^2)\right) > 0.5.$$

This implies that a sample from class $C$ is more likely to be classified as $A$ (non-$B$) than as $B$ under the Mahalanobis distance-based decision rule.

Therefore, when $\boldsymbol{\mu}_C$ is closer to $\boldsymbol{\mu}_A$ than to $\boldsymbol{\mu}_B$ in the Mahalanobis sense, using class $A$ as a non-$B$ background model leads to a correct classification (as non-$B$) with probability greater than $0.5$, making it a justified and effective open-set prior. $\square$

## A.4 VISUALIZATION

To better illustrate the superiority of AGE and its practical benefits, we provide t-SNE visualizations of both known and novel classes from the CUB-200 dataset. Fig. 7 depicts the feature distribution of known classes, where translucent gray points represent samples from both ground-truth novel classes and predicted novel classes. Samples with the same color correspond to the same predicted class. It is clear that our method not only accurately recognizes known classes but also yields reliable predictions. Fig. 6 presents the feature distribution of novel classes, with translucent gray points again indicating novel class samples. AGE demonstrates a pronounced advantage in clustering novel samples: instances from the same predicted class form compact clusters in the feature space while maintaining clear separations from other classes. These results highlight ability to effectively distinguish and structure novel class features without any prior category information, providing strong evidence of its effectiveness in novel class identification and modeling under the open-set recognition setting.

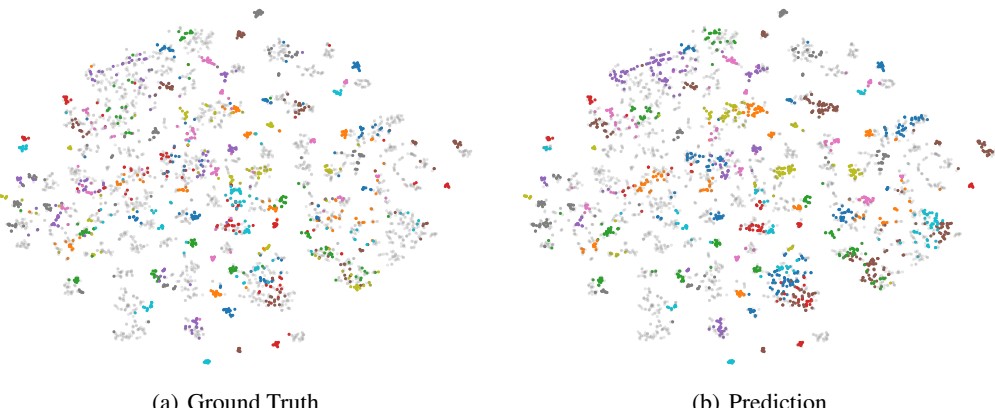

(a) Ground Truth                    (b) Prediction

Figure 5: The t-SNE visualization of known classes on the CUB-200 dataset.

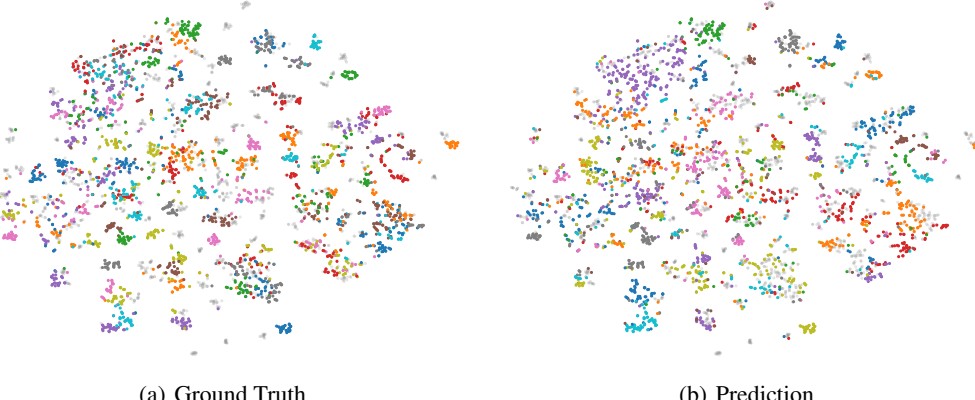

(a) Ground Truth                    (b) Prediction

Figure 6: The t-SNE visualization of novel classes on the CUB-200 dataset.

For an intuitive visualization of how categories emerge throughout the dynamic discovery process, we plot a series of t-SNE maps on the Pets dataset in a continual-learning style. (a) shows only the old classes; (b–d) illustrate the progressive emergence of new classes; (e) presents the final predictions for all newly discovered classes; (f) provides the corresponding ground truth.

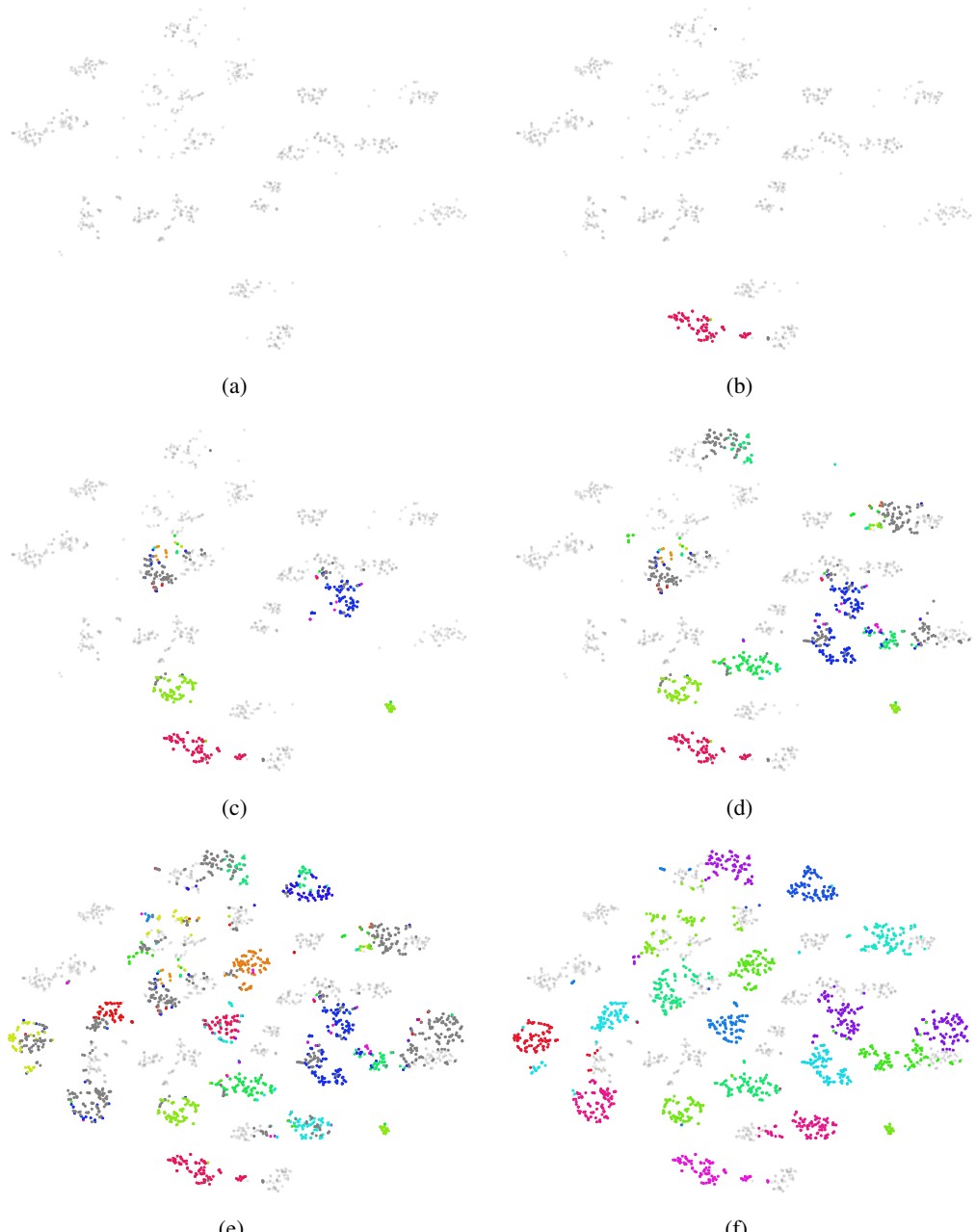

Figure 7: The t-SNE visualization on the Pets.

## A.5 Effetcs of $\beta$

In our experiments, we observed that using either a global threshold or a per-class threshold can lead to significant bias. A global threshold may overlook certain low-confidence classes, while per-class thresholds can be excessively noisy when the validation set is small. In Fig.8, we visualized the confidence distributions for several classes on CUB-200 along with thresholds under different $\beta$ values. The results show that the soft-thresholding strategy effectively mitigates such bias.

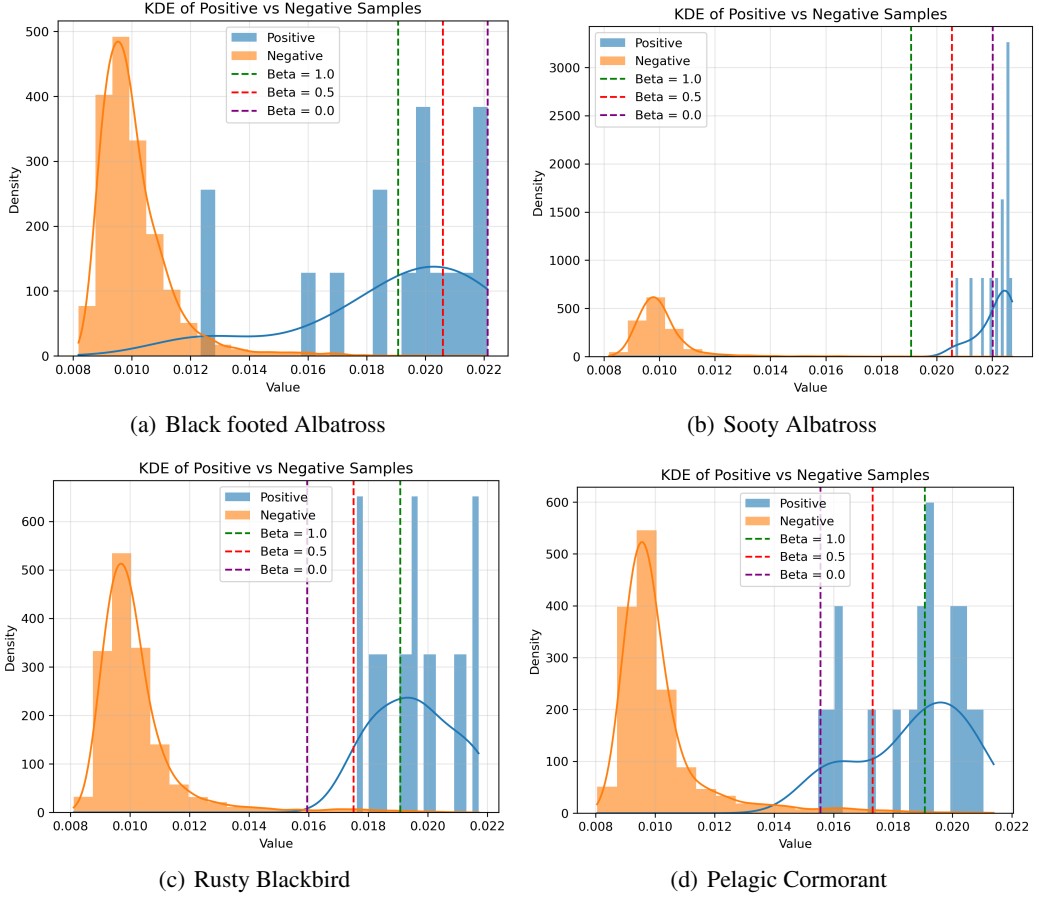

(a) Black footed Albatross      (b) Sooty Albatross

(c) Rusty Blackbird      (d) Pelagic Cormorant

Figure 8: Confidence distributions of selected classes on CUB-200

In Fig. 9, we summarize the impact of different values of the fusion coefficient $\beta$ on the performance of anomaly detection. The evaluation metric is the accuracy of OOD detection, defined as the proportion of samples correctly identified as either in-distribution or out-of-distribution.

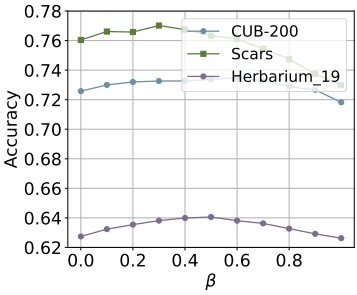

Figure 9: Performance with different $\beta$.

### A.6 PSEUDO-CODE

To improve clarity and ensure reproducibility, we provide the pseudocode of the proposed Adaptive Gaussian Expansion (AGE) inference procedure in Algorithm 1. This offers a concise summary of the key steps involved, particularly in the streaming setting where real-time category discovery is required.

---

**Algorithm 1** Adaptive Gaussian Expansion

---

**Input**: Validation set $D_V$, query stream $D_Q$, encoder $E$, projection head $H$, classifier $C$
**Parameter**: Threshold smoothing $\beta$, soft prior $s$, stability $\varepsilon$
**Output**: Predicted labels $\hat{\boldsymbol{Y}}_Q$

1: Extract features $\boldsymbol{f}_i = E(x_i)$ for all $x_i \in D_V$
2: Apply PCA to reduce feature dimensionality to $d$
3: Compute class-wise confidence means $\boldsymbol{\mu}_k$ and stds $\boldsymbol{\Sigma}_k$ on $D_V$
4: Compute thresholds: $\tau_k = \boldsymbol{\mu}_k - \boldsymbol{\Sigma}_k$, global $\tau_g = \frac{1}{|\boldsymbol{Y}_S|} \sum_k \tau_k$
5: Final threshold: $\tau'_k = \beta \cdot \tau_k + (1 - \beta) \cdot \tau_g$
6: Estimate $\boldsymbol{\mu}_k$, covariance $\boldsymbol{\Sigma}_k$ for each class using Ledoit–Wolf shrinkage
7: **for** each query sample $x_i \in D_Q$ **do**
8:     Extract and project feature $\boldsymbol{f}_i = \text{PCA}(E(x_i))$
9:     Compute prediction $p_i = \text{Softmax}(C(H(\boldsymbol{f}_i)))$
10:     **if** $\max(p_i) > \tau'_{\arg\max(p_i)}$ **then**
11:         Assign label $\hat{y}_i = \arg\max(p_i)$
12:     **else**
13:         Compute log-PDFs $\log \boldsymbol{p}_k$ using Eq. 13 for all novel clusters
14:         **if** predicted as known **then**
15:             Create new cluster with mean $\boldsymbol{f}_i$, covariance $\boldsymbol{\Sigma}_{\text{prior}}$
16:         **else**
17:             Assign to cluster $k^* = \arg\max \log \boldsymbol{p}_k$
18:             Update $\boldsymbol{\mu}_{k^*}$ and $\boldsymbol{\Sigma}_{k^*}$ with Ledoit–Wolf
19:         **end if**
20:     **end if**
21: **end for**
22: **return** predicted labels $\hat{\boldsymbol{Y}}_Q$

---

## A.7 STABILITY ANALYSIS OF AGE

To evaluate the stability of AGE, we conducted five independent runs and reported the mean and standard deviation of the results, as shown in Table 12. The results demonstrate that AGE consistently exhibits low variance across multiple datasets, indicating strong stability and robustness under different random initializations. AGE not only achieves higher average performance in most scenarios but also significantly reduces performance fluctuations, further validating its reliability and robustness in practical applications.

Table 12: Stability evaluation of AGE across five runs. Results are reported as mean ± standard deviation.

| Dataset | All | Old | New |
|---|---|---|---|
| CIFAR-100 | 60.24±0.17 | 75.63±0.46 | 29.46±0.85 |
| ImageNet-100 | 48.16±0.45 | 87.08±0.03 | 28.59±0.63 |
| CUB-200 | 46.20±0.52 | 61.57±0.40 | 38.51±0.67 |
| Scars | 35.01±0.10 | 63.05±0.51 | 21.47±0.48 |
| Pets | 61.58±1.77 | 66.60±0.01 | 58.94±2.71 |
| Herbarium19 | 30.93±0.25 | 54.57±0.08 | 18.20±0.40 |
| Fungi | 39.39±0.73 | 64.90±4.44 | 28.08±2.60 |
| Arachnida | 43.11±0.60 | 73.78±0.15 | 23.82±0.97 |
| Animalia | 44.33±0.44 | 66.78±0.18 | 35.03±0.58 |
| Mollusca | 43.23±0.40 | 69.95±0.04 | 29.01±0.62 |
| Food101 | 30.16±0.12 | 69.95±0.05 | 9.87±0.33 |

## A.8 COMPARISON WITH K-MEANS CLUSTERING

We further conducted comparative experiments with the classical clustering method K-means. Specifically, we directly performed clustering on all samples in the query set $D_Q$ during the inference stage. Since K-means relies on the global distribution of all samples during inference, it may have certain advantages in specific scenarios. We report the classification performance across multiple datasets and provide a comprehensive comparison with our proposed AGE method, as shown in Table 13.

As shown in the table, K-means achieves slightly better novel class recognition performance than AGE on some datasets, such as CIFAR-100 and ImageNet-100. However, this advantage mainly stems from its dependence on global distribution information, making it less suitable for online or incremental category discovery tasks. In contrast, AGE demonstrates stronger known-class retention and more stable novel-class discovery performance on certain datasets, particularly exhibiting significant advantages on fine-grained datasets such as Herbarium19 and Scars. This validates that AGE can achieve robust and generalizable category discovery without access to the global sample distribution.

Table 13: Comparison with the inductive K-means.

| Method | CIFAR-100 | | | ImageNet-100 | | | CUB-200 | | | Scars | | | Herbarium19 | | |
|---|---|---|---|---|---|---|---|---|---|---|---|---|---|---|---|
| | All | Old | New | All | Old | New | All | Old | New | All | Old | New | All | Old | New |
| K-means | 65.9 | 77.8 | 42.3 | 66.6 | 71.6 | 50 | 53.0 | 52.8 | 53.1 | 33.7 | 36.6 | 31.6 | 26.5 | 33.8 | 22.6 |
| AGE | 60.8 | 75.8 | 30.7 | 48.2 | 87.1 | 28.6 | 46.3 | 59.8 | 39.4 | 34.8 | 62.7 | 21.3 | 30.9 | 54.7 | 18.1 |

| Method | Fungi | | | Arachnida | | | Animalia | | | Mollusca | | | Pets | | |
|---|---|---|---|---|---|---|---|---|---|---|---|---|---|---|---|
| | All | Old | New | All | Old | New | All | Old | New | All | Old | New | All | Old | New |
| K-means | 40.0 | 55.9 | 33.0 | 45.3 | 65.5 | 32.6 | 44.9 | 53.8 | 41.2 | 44.7 | 68.3 | 32.2 | 69.0 | 52.8 | 77.6 |
| AGE (Ours) | 40.0 | 66.1 | 28.5 | 42.7 | 73.7 | 23.3 | 44.1 | 68.3 | 34.2 | 43.0 | 70.0 | 28.7 | 61.5 | 66.5 | 58.8 |

## A.9 TIME EFFICIENCY

To compare the time complexity of AGE with that of existing methods, we measured the inference time across multiple datasets, where the reported cost for AGE includes the threshold estimation on the validation set. As shown in Table 14, AGE incurs a slightly higher time cost than SMILE and PHE, yet achieves a substantial improvement in accuracy. It is worth noting that AGE does not require training on the validation set during the training phase, leading to a slightly reduced overall training time compared with prior methods.

Table 14: Time efficiency (seconds) of different methods.

|           | CUB-200 | Scars | Pets | ImageNet-100 | Herbarium9 | Fungi |
|-----------|---------|-------|------|--------------|------------|-------|
| SMILE     | 34      | 59    | 29   | 745          | 330        | 78    |
| PHE       | 34      | 60    | 29   | 747          | 331        | 78    |
| AGE (Ours)| 37      | 68    | 32   | 794          | 368        | 85    |

## A.10 EFFECTS OF TEST SAMPLE ORDERING

Since the model needs to retain information from previously observed test samples during inference, which inherently depends on their content, we compare the performance of AGE under different scenarios. In previous studies, test datasets are typically presented in an unshuffled order to better simulate real-world conditions, where novel classes often appear in succession—an observation aligned with the locality principle in computer memory access. Table 15 presents the impact of shuffling test samples on performance, showing that AGE exhibits low sensitivity to the temporal order of data, thereby highlighting its robustness.

Table 15: Performance with different ordering.

|         | CUB-200 |      |      | Pets |      |      |
|---------|---------|------|------|------|------|------|
|         | All     | Old  | New  | All  | Old  | New  |
| Order   | 46.3    | 59.8 | 39.4 | 61.5 | 66.5 | 58.8 |
| Shuffle | 46.4    | 62.8 | 38.1 | 62.1 | 66.6 | 59.8 |

## A.11 EFFECTS OF SLIDING-WINDOW SIZE

To examine the influence of sliding-window size on model performance, we evaluate two datasets where the effect is particularly evident. As shown in Table 16, accuracy improves steadily with increasing window size, suggesting that a larger window yields a more stable and reliable covariance estimation.

Table 16: Performance with different window size.

| Size | CUB-200 |      |      | Fungi |      |      |
|------|---------|------|------|-------|------|------|
|      | All     | Old  | New  | All   | Old  | New  |
| 2    | 32.7    | 63.2 | 18.0 | 37.3  | 67.4 | 23.9 |
| 4    | 34.6    | 63.7 | 20.6 | 38.0  | 66.6 | 25.4 |
| 8    | 35.2    | 63.1 | 21.7 | 38.6  | 67.3 | 25.6 |
| 16   | 35.2    | 62.7 | 21.9 | 39.5  | 67.3 | 27.3 |
| 32   | 35.3    | 63.1 | 21.9 | 39.9  | 67.3 | 27.9 |

## A.12 FURTHER DISCUSSION

Compared to existing methods, the advantages of AGE are not limited to its superior performance; more importantly, it demonstrates greater practicality in real-world applications. In approaches such as SMILE and PHE, identifying previously seen classes requires retrieving representative hash codes from the old class samples. However, due to the inherently sensitive nature of hash codes, these methods often suffer from low classification accuracy.

The proposed method effectively addresses the long-standing issue of suboptimal performance in feature-level clustering approaches. By using features as unique identifiers for samples during training, AGE eliminates the need for additional parameters or loss terms, thereby introducing no extra training computational overhead.

## A.13 FUTURE WORK

The current outlier detection module in AGE adopts a relatively simple thresholding strategy and does not leverage recent advances in the Open Set Recognition domain. Future research could explore deeper integration with state-of-the-art Open Set Recognition techniques to enhance outlier detection capabilities.

Moreover, under the constraint of streaming data, AGE has demonstrated even higher performance than K-means on certain datasets. This suggests the potential of integrating AGE into tasks such as Novel Category Discovery and Generalized Category Discovery.

Finally, we believe that incorporating language models to guide OCD can effectively alleviate the bias caused by the inherent invisibility of novel classes. Future work may consider exploring multi-modal approaches to further improve performance.

