# OpenReview forum: "Adaptive Gaussian Expansion for On-the-fly Category Discovery"
_ICLR.cc/2026/Conference — ICLR 2026 Poster_

### Official Review · Reviewer_BgKL · 2025-10-28

**Soundness:** 3
**Presentation:** 2
**Contribution:** 3
**Rating:** 6
**Confidence:** 5

**Summary:**

This paper studies the novel task of On-the-Fly Category Discovery (OCD), in which the model needs to perform real-time classification of potential future categories using prior knowledge. Previous methods suffer from severe overestimation of the number of categories. This paper first formally identifies a performance lower bound for the task, then decomposes OCD into two sub-tasks, i.e., Open-Set Recognition and a Fully Novel OCD setting. Specifically, this work proposes soft class thresholding and Adaptive Gaussian Expansion (AGE) for two tasks, respectively. Extensive experiments show the proposed AGE achieves remarkable performance.

**Strengths:**

1. This paper is well-motivated and easy to follow.
2. Some theoretical analyses are provided to validate the proposed insights.
3. The proposed method outperforms previous SOTA by a large margin on various datasets.
4. Detailed ablations are conducted to validate the validity of each component.

**Weaknesses:**

1. How to guarantee the estimated threshold using the validation set is precise and applicable to detect outliers/new classes in test samples? The authors should provide some explanations and experimental results of the first task, i.e., open-set recognition.
2. The writing and logic in Section 3.4 should be improved. For example, the lemma and proposition should be rearranged and interspersed with the viewpoint statement, rather than putting them all together at the end of Section 3.4.
3. According to my understanding, the novelty of AGE mainly lies in the inference time. I was wondering whether the method consumes more computations, so the comparison of inference time between the proposed method and previous works should be provided.

**Questions:**

See weakness. How to guarantee the estimated threshold could generalize to test-time samples?

---

> ### Author Response · Authors · 2025-11-21
>
> Thanks for the valuable feedback, which has helped further refine and improve AGE.
>
> ### **W1:**
> In our experiments, the validation set is sampled from the training set but excluded from training, and since the training and test sets follow the same distribution, the threshold estimated on the validation set remains valid for test samples. Ensuring reliable open-set detection under distribution shift is a long-standing challenge in OSR, and our method uses soft-confidence filtering mechanism.
> We visualize how different values of $\beta$ influence several representative classes.
> `See new submission Line.927`
>
> Recognizing that extreme feature-distribution shifts of old classes can cause fluctuations in novel-class accuracy (as noted by Reviewers NpEm and MoeU), we added experiments analyzing how different types of old-class noise impact novel-class performance.
>
> 1. **Masking all old-class samples**,
> 2. **Using the OSR result**,
> 3. **Injecting simulated old-class noise**.
>
> **Table: Performance under Different Noise Conditions.** *New: Accuracy on New Classes; Est.: Estimated Number of Classes.*
>
> |   | **CUB-200** |      | **Pets** |      | **Scars** |      |
> | - | ----------- | ---- | -------- | ---- | --------- | ---- |
> |   | New         | Est. | New      | Est. | New       | Est. |
> | 1 | 43.5        | 329  | 64.6     | 93   | 25.2      | 429  |
> | 2 | 43.9        | 401  | 64.2     | 147  | 25.1      | 634  |
> | 3 | 43.2        | 452  | 78.0     | 169  | 73.3      | 697  |
>
> We also now include results and explanations for the first task:
>
> 1. AUROC and FPR@TPR95 under different confidence definitions, `See new submission Line.502`
>
> **Table: Performance with different OSR methods.**
>
> | Method       | CUB-200 FPR@TPR95 | CUB-200 AUROC | Pets FPR@TPR95 | Pets AUROC |
> |--------------|-----------------|---------------|----------------|------------|
> | MSP          | 76.9            | 78.0          | 63.9           | 82.3       |
> | Energy Score | 77.7            | 77.9          | 61.5           | 84.3       |
> | MSP+         | 76.1            | 78.4          | 73.3           | 80.0       |
>
> 2. open-set detection accuracy using the proposed soft threshold, `See new submission Line.502`
>
>
> **Table: OSR Accuracy.** *Open Set Classification Accuracy (OSAcc) and Out-of-Distribution Accuracy (OODAcc)*
>
>
> |              | **OSAcc**       | **OODAcc**    | **OSAcc**      | **OODAcc** |
> |--------------|-----------------|---------------|----------------|------------|
> | MSP+Soft     | 75.4            | 74.8          | 70.1           | 60.9       |
>
>
> 3. threshold visualizations `See new submission Line.926`
> 4. the effect of varying thresholds on long-tailed datasets. `See new submission Line.961`
>
> ---
>
> ### **W2:**
> We appreciate the suggestion and have improved the structure and logical flow of Section 3.4 for clearer and more coherent presentation.
>
> ---
>
> ### **W3:**
> We acknowledge that inference time is critical in the OCD setting. In the previous submission, we reduced computation by caching inverse covariance matrices to avoid repeated matrix inversions during PDF evaluation, decreasing inference time significantly (CUB: 102s → 46s), though a gap remained relative to prior methods (OCD/PHE: 34s). In the revised version, we further optimized GPU-based inference, reducing runtime further (CUB: 46s → 37s), and we have updated the manuscript to summarize inference times across all datasets.
>
> **Table: Time efficiency (seconds) of different methods.**
>
> | Method         | CUB-200 | Scars | Pets | ImageNet-100 | Herbarium9 | Fungi |
> | -------------- | ------- | ----- | ---- | ------------ | ---------- | ----- |
> | SMILE          | 34      | 59    | 29   | 745          | 330        | 78    |
> | PHE            | 34      | 60    | 29   | 747          | 331        | 78    |
> | **AGE (Ours)** | 37      | 68    | 32   | 794          | 368        | 85    |

---

### Official Review · Reviewer_MoeU · 2025-10-30

**Soundness:** 2
**Presentation:** 3
**Contribution:** 3
**Rating:** 4
**Confidence:** 3

**Summary:**

This paper focuses on the On-the-Fly Category Discovery (OCD) task. The authors point out that existing OCD methods rely on hash-based encodings (e.g., PHE, SMILE), which lead to a serious overestimation of the number of categories and poor adaptability to downstream tasks. To address these issues, the authors propose: Establishing a theoretical lower bound, revealing that a closed-set classifier alone can achieve a certain level of performance; Reformulating the OCD task into two subtasks — Known Category Recognition (Open-Set Recognition, OSR) and Real-time Novel Category Discovery; Introducing the Adaptive Gaussian Expansion (AGE) framework, which enables adaptive expansion of class distributions through dynamic Gaussian modeling; Achieving significant performance improvements on multiple benchmark datasets.

**Strengths:**

1.	The paper is clearly written, well-structured, and effectively supported with figures and tables.
2.	The ablation study is relatively comprehensive, verifying the effects of factors such as soft covariance, threshold β, and PCA dimensionality on the model’s performance.

**Weaknesses:**

1.	The core characteristics of OCD lie in real-time streaming input and no requirement for global access to historical data. Although the AGE method claims to be “on-the-fly,” it actually depends on a pre-trained encoder, a validation set for estimating class-specific thresholds, and the need to update means and covariance matrices whenever a new sample arrives. Does such a design—requiring continual updates of historical statistics—truly satisfy the assumption of “real-time streaming discovery”? In a continuous data stream scenario, how can its time and memory complexity remain feasible as data grows linearly?
2.	Does AGE merely transfer the thresholding mechanism from the hash space to the Gaussian density space? Is there any theoretical or empirical evidence that it can control the number of discovered categories more accurately? If the threshold is chosen improperly, could it also lead to category fragmentation or under-segmentation?
3.	The paper claims to distinguish known and novel samples through soft-thresholding, yet this decision relies on fixed β-based confidence statistics. If the class distribution of the validation set differs significantly from that of the test set, can the threshold remain reliable? Does this mechanism truly embody the dynamic adaptivity required by OCD, or does it only work under static distributions?
4.	The paper lacks intuitive experimental visualizations to demonstrate how categories dynamically emerge during the on-the-fly discovery process.
5.	The literature review is incomplete, missing several recently published related works.

**Questions:**

Please see the weakness.

---

> ### Author Response · Authors · 2025-11-21
>
> We thank the reviewer for the detailed and helpful comments that improved the paper.
>
> ### **W1.**
>
> * We appreciate the insightful feedback. After revisiting the OCD formulation, we acknowledge that the original AGE implementation was not fully aligned with the OCD setting.
> * To address this, we introduce a *ixed-size per-class FIFO buffer, ensuring that memory grows with the *number of discovered classes*, not with the stream length. Here is effects of different window sizes.
>
> **Table: Performance with Different Window Sizes**
>
> | Size | CUB-200 All | CUB-200 Old | CUB-200 New | Fungi All | Fungi Old | Fungi New |
> |------|-------------|-------------|-------------|-----------|-----------|-----------|
> | 2    | 32.7        | 63.2        | 18.0        | 37.3      | 67.4      | 23.9      |
> | 4    | 34.6        | 63.7        | 20.6        | 38.0      | 66.6      | 25.4      |
> | 8    | 35.2        | 63.1        | 21.7        | 38.6      | 67.3      | 25.6      |
> | 16   | 35.2        | 62.7        | 21.9        | 39.5      | 67.3      | 27.3      |
> | 32   | 35.3        | 63.1        | 21.9        | 39.9      | 67.3      | 27.9      |
>
> * We further optimized GPU parallel inference and report the full computational overhead in the supplementary material. Experiments show that AGE incurs slightly higher time cost compared to SMILE and PHE, but achieves substantially higher accuracy.
>
> **Table: Time efficiency (seconds) of different methods.**
>
> | Method         | CUB-200 | Scars | Pets | ImageNet-100 | Herbarium9 | Fungi |
> | -------------- | ------- | ----- | ---- | ------------ | ---------- | ----- |
> | SMILE          | 34      | 59    | 29   | 745          | 330        | 78    |
> | PHE            | 34      | 60    | 29   | 747          | 331        | 78    |
> | **AGE (Ours)** | 37      | 68    | 32   | 794          | 368        | 85    |
>
>
> ---
>
> ### **W2.**
> * AGE does not merely transfer thresholding from hashing space to Gaussian space; the threshold is used only for open-set detection.
> * In contrast to PHE, whose threshold $d_{\max}$ derived from the Gilbert–Varshamov bound may cause fragmentation or insufficient splitting. AGE avoids dataset-specific tuning by operating in a generative, incremental Gaussian feature space.
> * Filtering out high-confidence old-class samples prevents them from forming spurious new clusters and controls overestimation of class numbers.
>
> To evaluate sensitivity to thresholding and old-class injection, we compare:
>
> 1. **Masking all old-class samples**,
> 2. **Using the original distribution**,
> 3. **Injecting simulated old-class noise**.
>
> **Table: Performance under Different Noise Conditions.** *New: Accuracy on New Classes; Est.: Estimated Number of Classes.*
>
> |   | **CUB-200** |      | **Pets** |      | **Scars** |      |
> | - | ----------- | ---- | -------- | ---- | --------- | ---- |
> |   | New         | Est. | New      | Est. | New       | Est. |
> | 1 | 43.5        | 329  | 64.6     | 93   | 25.2      | 429  |
> | 2 | 43.9        | 401  | 64.2     | 147  | 25.1      | 634  |
> | 3 | 43.2        | 452  | 78.0     | 169  | 73.3      | 697  |
>
> Results show that even when old-class samples enter the novel-class pool, **AGE remains robust** and performance does not degrade.
>
> ---
>
> ### **W3.**
> We thank the reviewer for the insightful comment. The design of $\beta$ is specifically intended to mitigate potential distributional discrepancies between the validation and test sets. Using either per-class thresholds or a global threshold may result in old classes being entirely absorbed into the novel-class region, or conversely, in novel classes being absorbed into old classes. To illustrate the effect of $\beta$ in the soft-thresholding computation, we visualize how different values of $\beta$ influence several representative classes.
> `See new submission Line.927`
>
> Regarding distributional differences in class composition, the validation set $D_V$ is derived from the labeled support set $D_S$, with known classes $\mathcal{Y}_S$ exactly matching those in the query set $D_Q$. Thus, for known-class samples, both their feature distributions and confidence scores align with the validation set, with no distributional shift except for changes in mixture proportion with novel classes. Even when the test stream contains only known or only novel classes, the static open-set detection partitions samples effectively, confirming AGE's robustness in dynamic environments.
>
> We argue that the dynamic adaptability of OCD mainly concerns handling the unknown nature of emerging classes, whereas open-set detection itself is a static procedure that does not require real-time expansion of the classifier.

---

> > ### Author Response · Authors · 2025-11-21
> >
> > ### **W4.**
> >
> > Because OCD processes data sample-by-sample, the incremental changes are too fine-grained to visualize meaningfully.
> > Following practices in continual learning, we provide stage-wise t-SNE plots on Pets in the supplementary material.
> > `See new submission Line.868`
> >
> > To help readers better understand how on-the-fly updates, GIF dynamic visualizations are included in the newly released codebase.
> > See [AGE](https://anonymous.4open.science/r/AGE-3B42/README.md).
> >
> > ---
> >
> > ### **W5.**
> >
> > As suggested, we add comparisons with the ICCV 2025 method **DiffGRE** and expand the description of PHE.
> >
> > | Method     | CUB-200 All | CUB-200 Old | CUB-200 New | Pets All | Pets Old | Pets New |
> > | ---------- | ----------- | ----------- | ----------- | -------- | -------- | -------- |
> > | DiffGRE    | 42.5        | 54.4        | 36.5        | 49.6     | 50.1     | 49.3     |
> > | AGE (Ours) | **46.3**    | **59.8**    | **39.4**    | **61.5** | **66.5** | **58.8** |
> >
> >
> > | Method     |Scars All | Scars Old | Scars New | Mollusca All | Mollusca Old | Mollusca New |
> > | ---------- | --------- | --------- | --------- | ------------ | ------------ | ------------ |
> > | DiffGRE    | 27.7      | 48.1      | 17.8      | 42.6         | 62.0         | 32.3         |
> > | AGE (Ours) | **34.8**  | **62.7**  | **21.3**  | **43.0**     | **70.0**     | **28.7**     |
> >
> > Results show AGE outperforms DiffGRE under the PHE+DiffGRE setting. Notably, DiffGRE relies on CLIP and generative models, whereas AGE achieves superior performance using only single-modality embeddings.

---

> > > ### Comment · Reviewer_MoeU · 2025-11-26
> > >
> > > Thank you for the authors’ response. My main concerns have been addressed, and I have therefore raised my score.

---

### Official Review · Reviewer_NpEm · 2025-10-30

**Soundness:** 3
**Presentation:** 4
**Contribution:** 3
**Rating:** 6
**Confidence:** 3

**Summary:**

The paper presents a novel framework called Adaptive Gaussian Expansion (AGE) for On-the-fly Category Discovery (OCD). The paper identify a performance lower bound for existing OCD methods and reformulate the task into two sub-problems: open-set recognition and fully novel OCD setting. AGE uses a probabilistic approach based on Gaussian distributions to model known and novel classes, enabling real-time classification and clustering without prior knowledge of class numbers. The method shows significant improvements over prior work across multiple benchmarks.

**Strengths:**

1. This paper establishes a theoretical lower bound for OCD, exposing the structural flaw of existing hashing methods that fail to fully exploit known-class information, and provides a rigorous benchmark for future work.
2. The task is decomposed into open-set recognition and novel class discovery: a threshold first filters known samples, then Gaussian clustering handles anomalies, cutting off most easy decisions early and reducing noise for the later stage.
3. AGE estimates per-class mean and covariance and explicitly computes membership probabilities via the multivariate Gaussian PDF, capturing intra-class shape and scatter and yielding finer uncertainty estimates along decision boundaries.

**Weaknesses:**

1.All samples falling below the confidence threshold are forwarded to AGE as novel candidates, yet the paper offers no mechanism to identify those that are merely low-confident members of known classes. Consequently, old-class noise is injected into the subsequent Gaussian estimation, biasing the covariance estimates and spawning spurious clusters.
2.Key parameters in the equations are reported only as bare values without physical interpretation or selection criteria, raising the barrier to comprehension. It is recommended that all parameters appearing in the formulas be thoroughly explained.
3.The paper lacks sufficient references to relevant research from the past three years. This undermines the persuasiveness of its argument regarding timeliness and innovation. It is recommended to supplement the study with comparative experiments and discussions involving SOTA methods published in recent years.

**Questions:**

1.Does the superior predictive performance on emerging categories contribute to the prediction of old categories?

---

> ### Author Response · Authors · 2025-11-21
>
> We are grateful to the reviewer for the constructive feedback and careful evaluation.
>
> ### **W1:**
> We agree that “old-class noise,” i.e., low-confidence samples from known classes that are misclassified as anomalies and forwarded to AGE, is a reasonable concern, as such noise may bias covariance estimation and lead to spurious new clusters. In general, low-confidence old-class noise is a known issue in OSR, and in this work, our class-specific soft-thresholding mechanism mitigates it to some extent. Nevertheless, we focus on addressing the proposed *Fully Novel OCD* setting.
>
> To investigate the potential negative impact of injecting old-class noise into subsequent Gaussian estimation for new-class discovery, we conducted an additional experiment. We report the accuracy of AGE in predicting new-class labels under the settings: (1) when all old-class samples are completely masked out, and (2) when extra old-class noise is injected. The additional noise is synthesized using the old-class mean and an expanded version of the global sample variance. This study helps us better understand how such noise may affect new-class discovery in our Fully Novel OCD scenario.
>
> 1. **Masking all old-class samples**,
> 2. **Using the original distribution**,
> 3. **Injecting simulated old-class noise**.
>
> **Table: Performance under Different Noise Conditions.** *New: Accuracy on New Classes; Est.: Estimated Number of Classes.*
>
> |   | **CUB-200** |      | **Pets** |      | **Scars** |      |
> | - | ----------- | ---- | -------- | ---- | --------- | ---- |
> |   | New         | Est. | New      | Est. | New       | Est. |
> | 1 | 43.5        | 329  | 64.6     | 93   | 25.2      | 429  |
> | 2 | 43.9        | 401  | 64.2     | 147  | 25.1      | 634  |
> | 3 | 43.2        | 452  | 78.0     | 169  | 73.3      | 697  |
>
> The accuracy differences across the three settings are minimal, indicating that AGE is highly robust to old-class noise and does not suffer noticeable degradation even when misrouted samples is introduced.
>
> ---
>
> ### **W2:**
> We have added explanations for several key hyperparameters:
>
>    - **$\alpha_k$**: $\alpha_k = \frac{s}{N_k + s}$ denotes the soft prior weight, introduced to prevent the covariance estimation from being excessively biased when the sample size is small, which could lead to a significant performance drop.  `See new submission Line.277`
>    - **$\varepsilon$**: The term $\varepsilon \boldsymbol{I}$ serves as a numerical stability factor. As $\varepsilon$ increases, the covariance approaches the identity matrix, causing the model to degenerate into a Euclidean distance-based model.  `See new submission Line.279`
>    - **$\beta$**: Represents the fusion ratio between class-specific and global thresholds. While a global threshold may overlook certain low-confidence classes, class-specific thresholds can become excessively noisy when the validation set is small.  `See new submission Line.229`
>    - **$s$**: The soft parameter $s$ was set to 2 and represents the relative importance of the prior covariance in subsequent estimation of class-specific covariances. `See new submission Line.370`
>
> ---
>
> ### **W3:**
> We have added a comparison with the ICCV2025 method **DiffGRE**, and also provide a introduction to PHE and DiffGRE. Results show that AGE outperforms DiffGRE on almost all datasets, based on the performance of PHE+DiffGRE. Notably, DiffGRE leverages CLIP and generative models to enhance performance, whereas AGE achieves superior results using embeddings from a single visual modality alone.
>
>
> | Method     | CUB-200 All | CUB-200 Old | CUB-200 New | Pets All | Pets Old | Pets New |
> | ---------- | ----------- | ----------- | ----------- | -------- | -------- | -------- |
> | DiffGRE    | 42.5        | 54.4        | 36.5        | 49.6     | 50.1     | 49.3     |
> | AGE (Ours) | **46.3**    | **59.8**    | **39.4**    | **61.5** | **66.5** | **58.8** |
>
>
> | Method     |Scars All | Scars Old | Scars New | Mollusca All | Mollusca Old | Mollusca New |
> | ---------- | --------- | --------- | --------- | ------------ | ------------ | ------------ |
> | DiffGRE    | 27.7      | 48.1      | 17.8      | 42.6         | 62.0         | 32.3         |
> | AGE (Ours) | **34.8**  | **62.7**  | **21.3**  | **43.0**     | **70.0**     | **28.7**     |
>
> Results show AGE outperforms DiffGRE under the PHE+DiffGRE setting. Notably, DiffGRE relies on CLIP and generative models, whereas **AGE achieves superior performance using only single-modality embeddings**.

---

> > ### Author Response · Authors · 2025-11-21
> >
> > ### **Q1:**
> > The superior performance of AGE is not merely due to accurate classification of old classes, but also stems from:
> > - providing accurate posterior probabilities through real-time estimation of mean and covariance, and
> > - decoupling new and old classes to mutually reinforce performance.
> >
> > Results on certain datasets (e.g., Pets, CUB-200) show that AGE’s new-class accuracy even exceeds the overall accuracy of other methods, and on Pets, it surpasses the old-class accuracy, indicating that new-class accuracy is not a burden but a significant improvement.
> >
> > To further validate this finding, we conducted an additional experiment in which old classes were masked, and accuracy was computed solely on the new classes.
> >
> > **Table: Performance with Different Noises**
> >
> > | Setting  | CUB-200 All | CUB-200 Old | CUB-200 New | Pets All | Pets Old | Pets New | Scars All | Scars Old | Scars New |
> > |----------|-------------|-------------|-------------|----------|----------|----------|-----------|-----------|-----------|
> > | Unmasked | 46.3        | 59.8        | 39.4        | 61.5     | 66.5     | 58.8     | 34.8      | 62.7      | 21.8      |
> > | Masked   | -           | -           | 43.9        | -        | -        | 64.2     | -         | -         | 25.1      |
> >
> > We also added t-SNE visualizations for novel classes on Pets. `See new submission Line.868`

---

### Official Review · Reviewer_hjpN · 2025-11-05

**Soundness:** 2
**Presentation:** 3
**Contribution:** 3
**Rating:** 4
**Confidence:** 3

**Summary:**

This paper analyzes OCD’s practical limits and establish a formal performance lower bound, motivating a reformulation into two sub-tasks: Open-Set Recognition and Fully Novel OCD. It introduces soft class thresholding to directly detect known classes, improving deployability, and propose Adaptive Gaussian Expansion (AGE), which models class PDFs to dynamically discover novel categories on the fly. Across multiple datasets, the proposed approach achieves state-of-the-art results.

**Strengths:**

1. The problem reframing is clear. Separating known-class retention (OSR) from novel class discovery is pragmatic and improves deployability.
2. The paper is clearly written and easy to follow.
3. The proposed method achieves strong empirical results.

**Weaknesses:**

1. The lemma 1 assumes shared covariance and whitening, which may be restrictive. The link from these assumptions to the actual AGE decision process (with smoothed per-class covariances and priors) could be tightened.
2. The soft class-wise threshold is somewhat heuristic. There is limited analysis of calibration, class imbalance, or alternative OSR baselines (e.g., energy score, MSP+temperature) for the thresholding module.
3. The paper describes deciding “falls within existing cluster” by maximum posterior but does not specify a principled threshold or DP concentration parameter mapping for new cluster instantiation; the “CRP-like” notion is qualitative rather than a formal nonparametric Bayesian update.
4. There're quite a lot of hyperparameters and the proposed method seems to be sensitive to hyperparameters. While some ablations are provided, key hyperparameters materially impact performance. Guidance or automatic selection strategies are limited.

**Questions:**

See weaknesses

---

> ### Author Response · Authors · 2025-11-21
>
> We appreciate the reviewer’s thoughtful feedback and valuable suggestions.
>
> ### **W1:**
> We appreciate your observation that the shared-covariance and whitening assumptions in Lemma 1 may be somewhat idealized. We would like to clarify that:
>
> - The primary purpose of **Lemma 1** and **Proposition 1** is to provide theoretical support for our key intuition: the distributions of known classes can serve as a reasonable prior for modeling unknown ones, rather than to serve as a direct mathematical formulation of the AGE algorithm.
> - In **AGE framework**, all class covariances are initialized using the global prior and smoothed via Eq. 11. This maintains an approximate form of the shared-covariance assumption in practice.
> - We have revised **Section 3.4** to strengthen the connection between these assumptions and the AGE decision process. `See new submission Line.332`
>
> ---
>
> ### **W2:**
> We thank the reviewer for the constructive feedback regarding the soft thresholding method:
>
> - The **soft thresholding method** is motivated by empirical observations across multiple datasets: using per-class thresholds or a global threshold may lead to either old classes being entirely absorbed into the novel-class region, or novel classes being absorbed into old classes. This results in complete loss of discovery ability for those novel classes.
> - In the revised version, we explicitly highlight this motivation and visualize, on the **CUB-200 dataset**, how different values of Beta affect representative classes.  `See new submission Line.927-956`
> - Regarding **class imbalance**, additional experiments on the **Herbarium_19** dataset demonstrate that the method remains effective under imbalanced class distributions.
> - Our primary goal is to clarify the redefinition of the **OCD task** and address the “Fully Novel OCD setting”. On the topic of **alternative OSR baselines**, we acknowledge this can be further improved. As noted in the “Future Work” section:
>   > Future research could explore deeper integration with state-of-the-art Open Set Recognition techniques to enhance outlier detection capabilities.
>
> - In the revised ablation study, we include comparisons using **energy scores** and **MSP with temperature scaling**, and report actual OSR accuracies. `See new submission Line.502-515 or below`
>
> We adopt MSP combined with the proposed soft-thresholding strategy as our OSR approach. In Table, we report the Open Set Classification Accuracy (OSAcc) and Out-of-Distribution Accuracy (OODAcc). We further investigate additional OSR methods, including Energy Score, MSP, and MSP+ for confidence estimation, and compare their AUROC and FPR@TPR95. The results indicate that using FPR95 induce a strong bias toward known classes, particularly on fine-grained datasets. MSP+ under sharpened logits becomes even more challenging for distinguishing OOD samples. It is worth noting that conventional OSR evaluation metrics are typically based on an optimal threshold, which ultimately requires selecting a reasonable threshold. The soft-thresholding strategy proposed in this work represents an effective attempt to apply various OSR methods in practice.
>
> **Table: Performance with different OSR methods.**
>
> | Method       | CUB-200 FPR@TPR95 | CUB-200 AUROC | Pets FPR@TPR95 | Pets AUROC |
> |--------------|-----------------|---------------|----------------|------------|
> | MSP          | 76.9            | 78.0          | 63.9           | 82.3       |
> | Energy Score | 77.7            | 77.9          | 61.5           | 84.3       |
> | MSP+         | 76.1            | 78.4          | 73.3           | 80.0       |
> |              | **OSAcc**       | **OODAcc**    | **OSAcc**      | **OODAcc** |
> | MSP+Soft     | 75.4            | 74.8          | 70.1           | 60.9       |
>
> ---
>
> ### **W3:**
> Regarding the “CRP-like” terminology. Thank you for raising this important point. Our use of the term “CRP-like” is intended to indicate the source of inspiration for AGE rather than to claim that AGE performs full nonparametric Bayesian inference.
>
> Traditional nonparametric Bayesian methods require access to all historical samples and multiple rounds of Gibbs sampling or variational optimization, with time and memory costs that grow with both the number of clusters and the data volume. In contrast, OCD operates in a strictly streaming, single-pass setting, where the model must make immediate decisions for each arriving sample, cannot revisit past data, and cannot perform global optimization. Under this constraint, a lightweight local-posterior decision rule better satisfies the real-time and single-channel memory requirements.
>
> In the next version, we will explicitly position AGE as a CRP-inspired method.

---

> ### Author Response · Authors · 2025-11-21
>
> ### **W4:**
> Regarding hyperparameter sensitivity:
>
> - AGE involves several hyperparameters, but experiments demonstrate **strong robustness**:
>   - Axis for $\varepsilon$ is magnified by 10×, showing <1% variation around 1e-5.
>   - Feature dimension is plotted on a logarithmic scale, with <2% change near the practically used 42 dimensions.
>   - Smoothing coefficient $s$ achieves optimal performance within 2–3, with adjacent values causing <2% difference.
> - These results indicate that the model is **robust to key hyperparameters**, requiring no fine-grained tuning to achieve stable performance. Notably, the same set of hyperparameters is used across all datasets, so no dataset-specific selection is necessary. We have also clarified the definitions of all hyperparameters in multiple sections of the manuscript to help readers understand their roles and make informed choices if desired. `See new submission Line.228, Line.277, Line.370`

---

> > ### Comment · Reviewer_hjpN · 2025-11-26
> > **Response to authors' rebuttal**
> >
> > Thanks for the authors’ rebuttal. My concerns have been addressed, and I have raised my score accordingly.

---

### Author Response · Authors · 2025-12-01
**Summary of Rebuttal**

Thank you for your careful reading.

---
The reviewers have generally acknowledged the motivation and the effectiveness of our paper.
We have summarized **all** their concerns along with our brief responses for your reference.
For specific details, please refer to the comments from each reviewer.

We would like to emphasize that before the data leakage, reviewers hjpN and MoeU had responded to our work.
We addressed their main concerns, and raised our score (before large-scale leak, the score reached **6666**).

**For convenience, we refer to Reviewer hjpN(4/3), NpEm(6/3), MoeU(4/3), and BgKL(6/5) as R1, R2, R3, and R4, respectively.**

---
### **Writing issues:**

**R1 W1:** The lemma 1 assumes shared covariance and whitening may be restrictive. The link from these assumptions to the actual AGE decision process could be tightened.

**R1 W3:** AGE does not specify a principled threshold or DP concentration parameter; the “CRP-like” notion is qualitative rather than a formal nonparametric Bayesian update.

**R4 W2:** The writing and logic in Section 3.4 should be improved.

> We clarified how Lemma 1 and Proposition 1 relate to AGE. As noted by Reviewer BgKL that they support intuition rather than define the method. We explained that “CRP-like” refers only to its inspiration under streaming constraints, and improved the clarity and structure of Section 3.4.

---
### **Further analysis of the first task:**

**R1 W2:** The soft class-wise threshold is somewhat heuristic. There is limited analysis of calibration, class imbalance, or alternative OSR baselines for the thresholding module.

**R4 W1:** Whether estimated threshold is precise to detect outliers/new classes in test samples? The authors should provide some explanations and experimental results of the first task.

> We clarified why the validation-set threshold is reliable, added analyses on old-class noise, expanded OSR results with AUROC, FPR95, OSAcc, and visualizations, and explained the motivation and effectiveness of soft thresholding across datasets, including long-tailed cases.

---
### **The effects of noise:**
**R2 W1:** Old-class noise is injected into the subsequent Gaussian estimation, biasing the covariance estimates and spawning spurious clusters.

**R3 W2:** Does AGE merely shift thresholding from hash space to Gaussian density space, and is there evidence it better controls the number of discovered categories? Could a mischosen threshold likewise cause fragmentation or under-segmentation?

**R3 W3:** If the class distribution of the validation set differs significantly from that of the test set, can the threshold remain reliable? Does this mechanism truly embody the dynamic adaptivity required by OCD, or does it only work under static distributions?

> We showed that AGE is robust to old-class noise, explained how soft-thresholding and $\beta$ prevent class absorption, noting that OCD handles emerging classes while open-set detection is static.


---
### **Computational efficiency issues:**
**R3 W1:** In a continuous data stream scenario, how can its time and memory complexity remain feasible as data grows linearly?

**R4 W3:** The comparison of inference time between the proposed method and previous works should be provided.

> We optimized the code to reduce both memory and time consumption and added comparative tables to illustrate performance differences.

---
### **Comparison with recent studies:**
**R3 W5:** The literature review is incomplete, lacking several recent and relevant studies.

**R2 W3:** We suggest adding additional experiments and discussions comparing the proposed method with recent SOTA approaches.

> We further compared our method with recent approaches, and the results demonstrate its superiority.

---
### **Hyperparameter issues:**
**R1 W4:** The proposed method seems to be sensitive to hyperparameters. Guidance or automatic selection strategies are limited.

**R2 W2:** It is recommended that all parameters appearing in the formulas be thoroughly explained.

> We clarified that the apparent high sensitivity to hyperparameters is actually due to the large axis ranges and provided further explanations for each hyperparameter.


---
### **Visualizations:**
**R3 W4:** The paper lacks intuitive experimental visualizations to demonstrate how categories dynamically emerge during the on-the-fly discovery process.

> We added visualizations.

---
### **Other Questions:**
**R2 Q1:** Does the superior predictive performance for emerging categories assist in predicting old categories?

> We included the accuracy on masked old classes, demonstrating that the performance on new classes remains equally superior.

---

### Meta-Review · Area_Chair_LmsZ · 2026-01-06

**Summary:**

The paper initially received mixed scores: 6, 6, 4, 4. The main concerns include: (1) Writing issues; (2) more results; (3) visualization; (4) comparison with more recent methods. The AC has carefully read the reviews and the rebuttal, and finds that the authors have largely addressed these concerns.

On the other hands, the AC finds that Reviewers hjpN and MoeU have raised their scores to 6 prior to the data leakage incident.

Given these considerations, the AC believes the main concerns have been addressed, and the reviewers are likely to revise their scores to be positive (e.g., 6, 6, 6, 6). The AC therefore recommends acceptance.

**Reviewer Concerns:**

Solved Concerns:

Reviewer hjpN: Clearer clarification,  more results.

Reviewer NpEm: Clearer clarification, more results, comparison with recent state-of-the-art.

Reviewer MoeU:  More results, clearer clarification,  comparison with recent state-of-the-art.

Reviewer BgKL:  More results; writing issue.

**Reviewer Scores:**

Reviewer hjpN and Reviewer MoeU will raise their scores from 4 to 6, as their concerns have been well-solved.

Reviewer NpEm and Reviewer BgKL will keep their original score of 6, as their concerns have been well-solved.

---

### Decision · Program_Chairs · 2026-01-26

Accept (Poster)